# Characterization and Toxicity Analysis of Lab-Created Respirable Coal Mine Dust from the Appalachians and Rocky Mountains Regions

Vanessa Salinas [1], Milton Das [2], Quiteria Jacquez [3], Alexandra Camacho [3], Katherine Zychowski [3], Mark Hovingh [4], Alexander Medina [1], Gayan Rubasinghege [2], Mohammad Rezaee [4], Jonas Baltrusaitis [5], Neal Fairley [6] and Pedram Roghanchi [1,*]

1   Department of Mineral Engineering, New Mexico Institute of Mining and Technology, Socorro, NM 87801, USA; vanessapaola.salinastorres@student.nmt.edu (V.S.); alexander.medinagomez@student.nmt.edu (A.M.)
2   Department of Chemistry, New Mexico Institute of Mining and Technology, Socorro, NM 87801, USA; milton.das@student.nmt.edu (M.D.); gayan.rubasinghege@nmt.edu (G.R.)
3   College of Nursing, University of New Mexico-Health Sciences Center, Albuquerque, NM 87106, USA; qsanchez@salud.unm.edu (Q.J.); acamacho40@gmail.com (A.C.); kzychowski@salud.unm.edu (K.Z.)
4   Department of Energy and Mineral Engineering, The Pennsylvania State University, University Park, PA 16802, USA; mah6364@psu.edu (M.H.); m.rezaee@psu.edu (M.R.)
5   Department of Chemical and Biomolecular Engineering, Lehigh University, Bethlehem, PA 18015, USA; job314@lehigh.edu
6   Casa Software Ltd., Bay House, 5 Grosvenor Terrace, Teignmouth, Devon TQ14 8NE, UK; neal@casaxps.com
*   Correspondence: pedram.roghanchi@nmt.edu; Tel.: +1-575-835-5273

**Abstract:** Coal mine workers are continuously exposed to respirable coal mine dust (RCMD) in workplaces, causing severe lung diseases. RCMD characteristics and their relations with dust toxicity need further research to understand the adverse exposure effects to RCMD. The geographic clustering of coal workers' pneumoconiosis (CWP) suggests that RCMD in the Appalachian region may exhibit more toxicity than other geographic regions such as the Rocky Mountains. This study investigates the RCMD characteristics and toxicity based on geographic location. Dissolution experiments in simulated lung fluids (SLFs) and in vitro responses were conducted to determine the toxicity level of samples collected from five mines in the Rocky Mountains and Appalachian regions. Dust characteristics were investigated using Fourier-transform infrared spectroscopy, scanning electron microscopy, the BET method, total microwave digestion, X-ray diffraction, and X-ray photoelectron spectroscopy. Inductively coupled plasma mass spectrometry was conducted to determine the concentration of metals dissolved in the SLFs. Finer particle sizes and higher mineral and elemental contents were found in samples from the Appalachian regions. Si, Al, Fe, Cu, Sr, and Pb were found in dissolution experiments, but no trends were found indicating higher dissolutions in the Appalachian region. In vitro studies indicated a proinflammatory response in epithelial and macrophage cells, suggesting their possible participation in pneumoconiosis and lung diseases development.

**Keywords:** RCMD; respirable dust characteristics; simulated lung fluids; in vitro toxicity studies

## 1. Introduction

Respirable coal mine dust (RCMD) refers to the mixture of airborne particles present in the air of surface and underground coal mines coming from different sources such as the rock breakage, intake air, rock dusting, diesel equipment, and any activity involving abrasion [1–3]. These particles are small enough to reach the deepest region of the human lungs, causing damages to the lung tissues [1,4–6]. Several researchers have described RCMD as those particles that are smaller than 10 µm and/or with a mean particle size of

4 μm (aerodynamic diameter) [1,4,5,7], but some others restrict the respirable fraction only to the one smaller than 4 μm [2,6].

Coal mine workers are frequently at risk for RCMD inhalation. A portion of dust may be retained by the mask (if worn properly) and in the upper part of the respiratory system. Large particles may be eliminated via mucocilliary clearance [8]. Smaller particles penetrate deep into the lungs and are deposited in the alveolar region. A portion of the inhaled particles stay in the alveoli since the body's own defense mechanisms are not able to expel them [7,9]. Long term exposure to RCMD can lead to pulmonary diseases such as coal workers' pneumoconiosis (CWP), mixed dust pneumoconiosis, dust-related diffuse fibrosis (DDF), progressive massive fibrosis (PMF), emphysema, chronic bronchitis, and silicosis. All these diseases are irreversible and may result in serious lung injuries and death [5].

During the last decades, tremendous efforts have been made to reduce miners' exposure to RCMD. However, lung diseases caused by RCMD still remain a major concern in the coal mining industry. In the early 2000s, an increase in the prevalence and severity of CWP, especially in central Appalachia, was observed [5,10,11]. The data indicated that coal miners in the Appalachia and Interior regions are at a higher risk of CWP prevalence in comparison with the Western region [12].

The number of CWP cases in the US by state and county (from 1986 to 2018) shows hot spot areas in the Appalachian region with a higher number of cases in West Virginia, Kentucky, Virginia, and Pennsylvania [12]. Additionally, an analysis of the chest radiographs of underground coal miners reported from 1996 to 2002 found CWP in 3% of the miners evaluated. From the miners that showed CWP, 35.4% presented rapidly progressive CWP and 14.8% evidenced PMF. Furthermore, a proportion between 61.5% and 80% of evaluated miners with rapidly progressive CWP was found in different counties in West Virginia, Kentucky, Virginia, and Pennsylvania, suggesting a cluster along the Appalachian region [13]. Similar observations that suggest a higher incidence of lung diseases in the Appalachian region have been obtained by NIOSH when examining chest radiographs from the Coal Workers' Health Surveillance Program (CWHSP) [5].

A study by Rahimi [14] investigated the geographic location of an underground mine as a contributing factor in RCMD and respirable crystalline silica (RCS) concentrations. The study used the Mine Safety and Health Administration (MSHA) database between 1989 and 2018, dividing the data into Appalachia, Interior, and Western regions. This study determined that geographic location is a contributing factor for RCMD concentrations in underground mines (Interior vs. Western) and RCS concentration for surface and underground mines (Appalachia vs. Western). Further research in the key differences in coal dust characteristics from different regions was recommended [14].

The higher rates of CWP cases in the Appalachian region have been linked to the thinner seams usually found in the mines from this region [11,12]. Several studies have demonstrated a substantially higher RCS content in dust samples from thin coal seam mines as a result of host rock cutting during coal extraction [1,11]. Silica was classified as a "Class I Human Carcinogen" by the International Agency for Research on Cancer and has been pointed to as the main cause of silicosis [5,10]. Long term exposure to RCS is also known to cause chronic obstructive pulmonary diseases (COPD) and lung cancer [15].

Although several studies have focused on the RCMD characterization [7,11,16,17], investigation of the toxicity of RCMD based on the elements and their influence on the inflammatory response of the lungs remains scarce. However, investigation of RCMD characteristics and the level of toxicity will significantly help to achieve a better understanding of the true reasons for the higher prevalence of lung diseases in the Appalachian region.

This study aims to characterize and analyze the toxicity of dust particles from different regions in the US. The samples for this study were collected from five mines located in the Appalachian region, the area of interest, and in the Rocky Mountains, a different basin, and coal occurrence zone in the US, this was to have two regions for comparison. The toxicity analysis of dust particles was conducted based on the elements that dissolve

in simulate lung fluids (SLF) and the inflammatory response they produce. For this purpose, dissolution experiments in simulated lung fluids (SLF) and in vitro responses were conducted. The characterization of dust samples was performed using Fourier transform infrared spectroscopy (FTIR), a scanning electron microscope (SEM), the BET method, microwave total digestion, X-ray diffraction (XRD), and X-ray photoelectron spectroscopy (XPS) to obtain the functional groups, particle size distribution, specific surface area, elemental content, mineral composition, and surface composition of the samples, respectively. The dissolution experiment exposed dust samples to two SLFs (Gamble's solution and artificial lysosomal fluid) and the concentrations of the metals dissolved after 24 h were obtained using inductively coupled plasma mass spectrometry (ICP-MS). For in vitro responses, HL-60, A549, and THP-1 cells were exposed to different concentrations of RCMD to evaluate the change in protein expression. Results were analyzed to find trends related to geographic location in order to link the characteristics and toxicity of dust samples to the higher incidence of lung diseases in the Appalachian region.

## 2. Materials and Methods

### 2.1. Bulk Sample Collection and Preparation

Bulk samples used in this study were collected from 5 mines: 3 mines from the Appalachian region and 2 from the Rocky Mountains. The purpose of bulk sample collection was to create a representative collection of materials in a mine that are aerosolized and subsequently inhaled by the miner during their shift. Samples were collected from the coal seam of the working face in order to collect material that closely matches the excavated coal rock.

Mines 1 and 2 correspond to the samples collected from the West regions, and Mines 3, 4, and 5 to the ones collected from the Appalachian region. It is important to mention that Mine 1 is a surface mine, and the rest are from underground room and pillar mines. The descriptions of the mines' location, type, and methods are shown in Table 1.

**Table 1.** Overall information of the samples collected for this research.

| Region | Mine ID | Type | Mining Method | Coal Rank | Seam Hieght | Ash (%) | Volatile Matter (%) | Sulfur (%) |
|---|---|---|---|---|---|---|---|---|
| Rocky Mountains | Mine 1 | S * | Open cut | Sub-bituminous | Low seam | 16.90 | 32.00 | 1.08 |
| | Mine 2 | UG ** | Room and pillar | Bituminous | High seam | 11.64 | 33.39 | 0.55 |
| Appalachians | Mine 3 | UG | Room and pillar | Sub-bituminous | Low seam | 5.85 | 16.26 | 1.07 |
| | Mine 4 | UG | Room and pillar | Sub-bituminous | Low seam | 11.28 | 15.43 | 1.17 |
| | Mine 5 | UG | Room and pillar | Bituminous | High seam | N/A *** | N/A | N/A |

* S: Surface; ** UG: Underground; *** N/A: Not available.

For the sample preparation, bulk sample size was initially reduced with a jaw crusher or mortar and pestle, depending on the initial size of the sample, until reaching 100% passing the U.S.A. standard sieve No. 6 (3.35 mm, ASTM E11). The resulting sample was ground with a 755RMV jar mill of 9.5 inches of diameter and 8.5 in of height, using media of zirconia $1/2'' \times 1/2''$ radius end cylinder, magnesia stabilized. First, the material was ground for 6 h, and sieved using the U.S.A. standard sieve No. 120 (opening of 125 μm) in order to remove the big particles that the mill was unable to reduce. Then, the material passing the sieve was ground for 6 more hours. To obtain a larger fraction of particles smaller than 10 μm, the material was ground additionally with a RETSCH XRD-Mill McCrone, which preserves the structure of the coal samples in the reduction process. The grinding was conducted for 5 min in a 4-step process, loading only 2 mL of sample per round, and using agate as media. Finally, the samples less than 10 μm fraction (mass mean aerodynamic diameter) were separated using a next-generation cascade impactor (NGI, model 170 NGI, MSP Corporation, Shoreview, MN, USA) with an attached aerolizer and gravimetric stages. Dust samples were weighed and loaded into hydroxypropyl methylcellulose capsules and drawn through an induction port using a pump Copley Scientific (Copley Scientific,

Nottingham, UK) operated at a flow rate of 60 L/min for 4 s. The fractions less than 10 μm were collected from multiple stages and used for further studies.

### 2.2. Dust Characterization

Several characterization techniques were used to study the initial conditions and different characteristics of the dust particles, such as particle size distribution, specific surface area, functional groups, and mineral, elemental, and surface composition. Later, results were compared to observe if they were any differences or trends among the different regions studied.

### 2.2.1. SEM

A NOVA-Nano-SEM-450 from the Center for Integrated Nanotechnologies (CINT) was used to collect SEM images to verify the particle size from the previous separation and to obtain the particle size distribution in the samples. The images were analyzed for the particle size using the software package ImageJ. An average of 80 particles were measured in its width and length to plot histograms of the particle sizes for both dimensions. Particulate matter (PM) fractions were calculated using the width of the particles.

### 2.2.2. XRD

A PANalytical X'Pert Pro diffractometer (PANalytical B.V., Almelo, The Netherlands), equipped with a Cu Kα source and a fixed divergence silt of 0.25°, was used to determine the mineral components in the samples. The analyses were conducted in a 40 min program with a continuous scan of 0.008° step size, operating at 45 kV and 40 mA, with a 2θ scan range from 5° to 70° and a scanning time of 40 s/step. Raw data were analyzed with HighScore Plus software (Malvern Panalytical Ltd., Malvern, UK) with a minimum significance of 2 and restricted to the dataset of minerals.

### 2.2.3. Total Microwave Digestion

A high-performance microwave system from Milestone (model ETHOS UP, Shelton, CT, USA), was used for the total digestion. The system is equipped with infrared and direct contactless temperature sensors and two 950-Watt magnetrons for a total power of 1900 W, which can operate up to 230 °C and 100 bar. TFM vessels with high resistant PEEK shields were used in a SK-15 rotor. The method used to digest the coal samples was the SK-PE-017, suggested by the manufacturer, which consisted in a two-step digestion for up to 200 mg of sample. First step used 10 mL of $HNO_3$ with a program of 190 °C and 800 W/1200 W of power (for 3 or less vessels/4–8 vessels), using 10 min to reach the temperature and 15 min of standing time. Second step used 2 mL of HF with a program of 230 °C using 20 min to reach the temperature and 15 min of standing time. For both steps, 800 W were used when operating with 3 or less vessels and 1200 W when operating 4–8 vessels. The standard reference material (SRM) CLB-1 from the USGS was used for quality control [18]. After digestion, the resulting solution was filtered and dissolved with RO water up to complete 50 mL of solution. Finally, elemental content was measured with an inductively coupled plasma mass spectrometer (ICP-MS), using a 1:10 dilution.

ICP-MS used was an Agilent Technologies model 7900. For the samples from the microwave digestion, a HF-resistant set-up was used, equipped with a sapphire torch, a PFA spray chamber capable to cool down up to 2 °C, and a PFA Scott-type concentric nebulizer. The following 29 elements were analyzed with the method: Li, Be, Mg, Al, Si, K, Ca, Ti, V, Cr, Mn, Fe, Co, Ni, Cu, Zn, As, Se, Sr, Mo, Ag, Cd, Sn, Sb, Ba, Tl, Pb, Th, and U. Limit of detection of the method for Mg, K, Ca, and Fe was around 100 ppb, for Mo 2 ppb, and for the rest of the elements 1 ppb.

### 2.2.4. BET

The surface area and the micro-pore of the dust samples were analyzed in a 92 points (52 points $N_2$ adsorption and 40 points desorption) Brunauer–Emmet–Teller (BET) isotherm using a Quantachrome Autosorb-1 instrument. Samples were outgassed for over 24 h at 150 °C before the analysis. The surface area was obtained using 7 $N_2$ adsorption points from the linear region of the isotherm. The pore size analysis was performed using Quenched Solid Density Functional Theory (QSDFT), which considers the surface roughness and heterogeneity and offers a reliable pore size analysis for any unknown carbon sample.

### 2.2.5. XPS

X-ray photoelectron spectroscopy (XPS) data were recorded using SPECS instrument (SPECS, Berlin, Germany) equipped with Phoibos 1D-DLD hemispherical electron energy analyzer and 0.3 mm entrance aperture, XR50MF aluminum K-$\alpha$ X-ray source operating at 100 W with m-FOCUS 600 X-ray monochromator, and in ultra-high vacuum (UHV) mode. Survey spectra were acquired using 100 eV pass energy, while high-resolution scans were acquired using pass energy of 20 eV. Low-energy electrons were used to neutralize the charge. Data processing was performed using CasaXPS software [19]. Charge calibration was performed using a 284.4 eV graphitic peak [20]. C1s spectrum peak fitted with synthetic components based on the literature data [21]. Elemental quantification was performed using relative sensitivity factors of 2.93 for O1s and 1.0 for C1s with correction for the instrument transmission/escape depth applied.

### 2.2.6. FTIR

FTIR spectra of the dust samples were collected using a Nicolet iS50 series FTIR (Thermo Scientific™, Waltham, MA, USA) equipped with Ge-ATR crystal. The sample was prepared in a small centrifuge tube. Approximately 10 mg of coal dust was sonicated for 20 min in 1 mL of isopropyl alcohol (IPA) and transferred to ATR crystal followed by air drying, which leaves a thin layer of dust on the ATR crystal. Collected spectra were processed against the spectra of bare crystal and only the processed spectra were reported. For each sample, FTIR analysis was performed for the parent dust sample before any dissolution experiment as well as after the dissolution in both GS and ALF. For this purpose, we recollected undissolved dust after the dissolution experiment using filtration and a hot plate to dry it.

### 2.3. Toxicity Analysis

Toxicity of dust samples was assessed throughout dissolution experiments to determine the metals dissolving in SLF and in vitro analysis to examine inflammatory response.

### 2.3.1. Dissolution Experiment

Dissolution experiments in SLFs were carried out to analyze the elements that dissolve from the dust samples and were in contact with the SLF for 24 h. Gamble's solution (GS) and artificial lysosomal fluids (ALF) were used as simulated lung fluids following the preparation shown in Table S1 [9,22]. GS simulated the pulmonary surfactants secreted by cells in the interstices of the lungs, and ALF simulates the acid fluid in macrophages that is in charge of trap and eliminate foreign bodies [8]. To simulate body conditions, experiments were run in a dark room, double jacketed bottles were used to circulate water at 37 °C and keep temperature constant, and the SLF were oxygenated for 5 min at a rate of 5 L/min before starting the experiments. Constant stirring at 1000 rpm was provided during the experiments. Triplicates were conducted for each sample in each SLF. Totals of 100 mL of SLF and 20 mg of coal sample were used in each trial. Sample aliquots of 1.5 mL were collected from the bottles before adding the coal to the SLF, and right after adding the coal to set the 0 h. Then, samples were collected at 1 h, 3 h, 6 h, 9 h, 12 h, 18 h, and 24 h. After collection, SLF samples were centrifuged, filtered, and stored in a freezer until analyzed. The same ICP-MS with a standard set-up was used to determine the elemental

content in the SLF dissolution. Standard set-up used a quartz torch, quartz spray chamber, and borosilicate glass Scott-type concentric nebulizer.

### 2.3.2. In Vitro Analysis

The first step consisted in the cell culture. THP-1(monocytic/macrophage), HL-60 (neutrophilic), and A549 (lung epithelial) cells (ATCC, Manassas, VA, USA) were incubated at 37OC in complete media according to manufacturer's instructions. Cells were monitored for confluence and appropriately passaged periodically. THP-1 and HL-60 cells were seeded ($2.0 \times 10^5$ cells per well) in 24 well plates and were then differentiated using 1.25% DMSO in media over the course of 5 days. A549 cells were similarly seeded at $2.0 \times 10^5$ cells per well and did not require differentiation.

For PM in vitro exposures, HL-60 cells were treated with a low (5 μg/mL), medium (10 μg/mL), and high (20 μg/mL) concentration of previously fractionated PM10. A549 and THP-1 cells were exposed to a low (10 μg/mL) and high (100 μg/mL) concentration of PM10. Each of these cell lines were exposed to PM10 for 4 h, and each PM-treatment was run in either duplicate or triplicate technical replicates. Supernatants were then collected for further analysis.

Proinflammatory Panel 1 (Human) Kit V-Plex (K15049D-1, Meso Scale Diagnostics, Rockville, MD, USA) was used to assess cytokine expression in the HL-60 and A549 cells from PM10 exposures. The following cytokines were evaluated for HL-60 and A549 cells: IFN-$\gamma$, IL-1$\beta$, IL-2, IL-4, IL-6, IL-8, IL-10, IL-12p70, IL-13, and TNF-$\alpha$. Cytokine Panel 1 (Human) Kit V-Plex (K15050D-1, Meso Scale Diagnostics, Rockville, MD, USA) was used to assess cytokine expression, including GM-CSF, IL-1$\alpha$, IL-5, IL-7, IL-12/IL-23p40, IL-15, IL-16, IL-17A, TNF-$\beta$, and VEGF-A in THP-1 cells. Meso Scale plates were run according to manufacturer's instructions. Briefly, supernatant was collected and pipetted onto plates. These plates were incubated with gentle shaking for 2 h at room temperature. Plates were washed three times with buffer solution. Detection antibodies were added to the wells and reacted at room temperature for 1 h. Read buffer was added to each well and plates were analyzed on an Meso Scale Discovery QuickPlex SQ instrument (Meso Scale Diagnostics, Rockville, MD, USA). Discovery Workbench software was used to calculate cytokine concentrations based on each cytokine standard curve. Change in protein expression was evaluated by the following equation: Log = exposed cell concentration/control and plotted according to each dust sample.

## 3. Results and Discussion

### 3.1. Particle Size Distribution

From the SEM analysis, it was verified that all samples were under 10 μm. Only few particles were outside the range, representing in its maximum 1% of the total number of particles analyzed. Histograms with the particle size distribution were built by the number of particles. Table 2 shows the information obtained from the SEM analysis, including the mean width and length of the particles, and the percentage fraction of the particulate matter (PM) less than 1 μm (PM1), 2.5 μm (PM2.5), 4 μm (PM4), and 10 μm (PM10).

Sub-micron and supra-micron fractions were extracted from the particle size information. It was observed that the majority of the particles were in supra-micron size. Only Mine 4 and Mine 5 had sub-micron particles, representing the 4% and 7% of the particles counted, respectively. However, it is important to mention that with the resolution of the images obtained, it was difficult to measure individual particles less than 1 μm or differentiate them from the structural layers in the coal, which may have led to an underestimating of the sub-micron fraction.

Mines from the Appalachian region had both a mean width and length smaller than samples from the Rocky Mountains, which were approximately 20% coarser. Considering that all the samples were prepared in the same way, the differences in the particle size distributions are attributed to the specific characteristics of each sample, such as the hardness and mineral contents. Additionally, significant differences in the percentages of

PM4 and PM2.5 were found, indicating that the overall samples from the Appalachian region contain a higher number of finer particles.

Angular edges may influence considerably the interaction of the particles with lung tissue, increasing the inflammation [1]. In the images (Figure 1), even if the particle shape was not extensively analyzed, it can be observed that overall, particles are not completely sharp. The samples from the different mines, all showed a transitional particle shape, according to the qualitative shape classification used by Sellaro, et al., 2015, which are particles in-between an angular and a rounded shape [7]. All samples were prepared following the same procedure, which may have provided similar particle shapes within the mines. In addition, the extensive hours of grinding may have rounded the edges of the particles, which is not necessarily the particle shape produced by the specific equipment and methods used in each mine. Since there are not significant differences in the particle shapes between mines and these ones, they may not be representative, and they were not considered as an influential factor for this study.

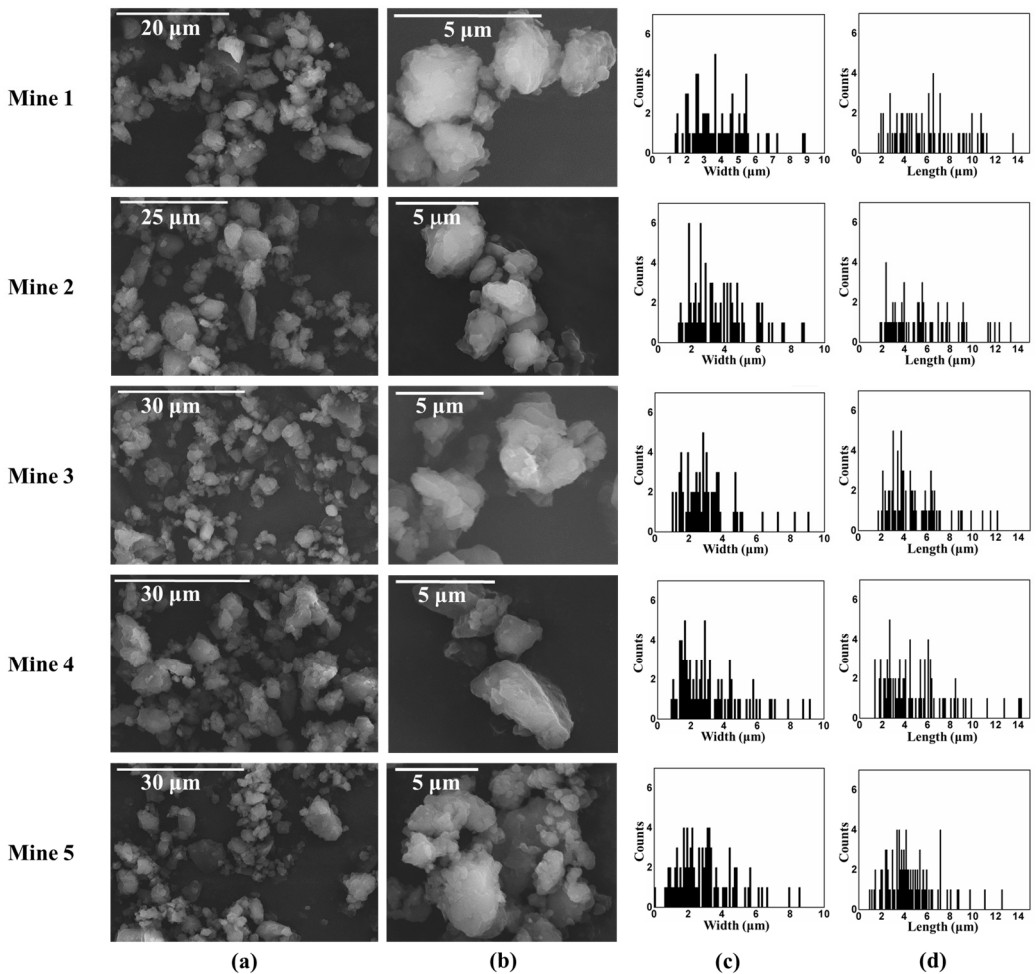

**Figure 1.** SEM results from the 5 mines. (**a**) SEM images. (**b**) Zoomed-in SEM images. (**c**) Particle size distribution based on width. (**d**) Particle size distribution based on length.

**Table 2.** Mean width and length of the particles and percentages of different particle factions.

| Mine ID | Mean Width (μm) | Mean Length (μm) | PM1 (%) | PM2.5 (%) | PM4 (%) | PM10 (%) |
|---|---|---|---|---|---|---|
| Mine 1 | 3.86 ± 1.62 | 6.03 ± 2.80 | 0 | 18 | 58 | 100 |
| Mine 2 | 3.81 ± 2.06 | 5.70 ± 3.20 | 0 | 28 | 62 | 99 |
| Mine 3 | 3.15 ± 1.76 | 4.82 ± 2.26 | 0 | 39 | 83 | 99 |
| Mine 4 | 3.24 ± 2.00 | 5.07 ± 3.10 | 4 | 44 | 73 | 99 |
| Mine 5 | 2.95 ± 1.65 | 4.43 ± 2.11 | 7 | 46 | 77 | 100 |

### 3.2. Mineral Composition

The mineral phases present in the coal samples are reported in Table 3. The XRD patterns obtained (Figure 2) showed that the five mines studied have in common the presence of quartz and kaolinite as expected. Additionally, in all the samples except for Mine 5, pyrite was observed. These three minerals showed to be the main mineral components in the coal samples. It was also observed small peaks for siderite and calcite in Mine 4 and Mine 5, respectively. The limitation of the method is that it is difficult to state the exact amount of each mineral, but relative abundances can be compared.

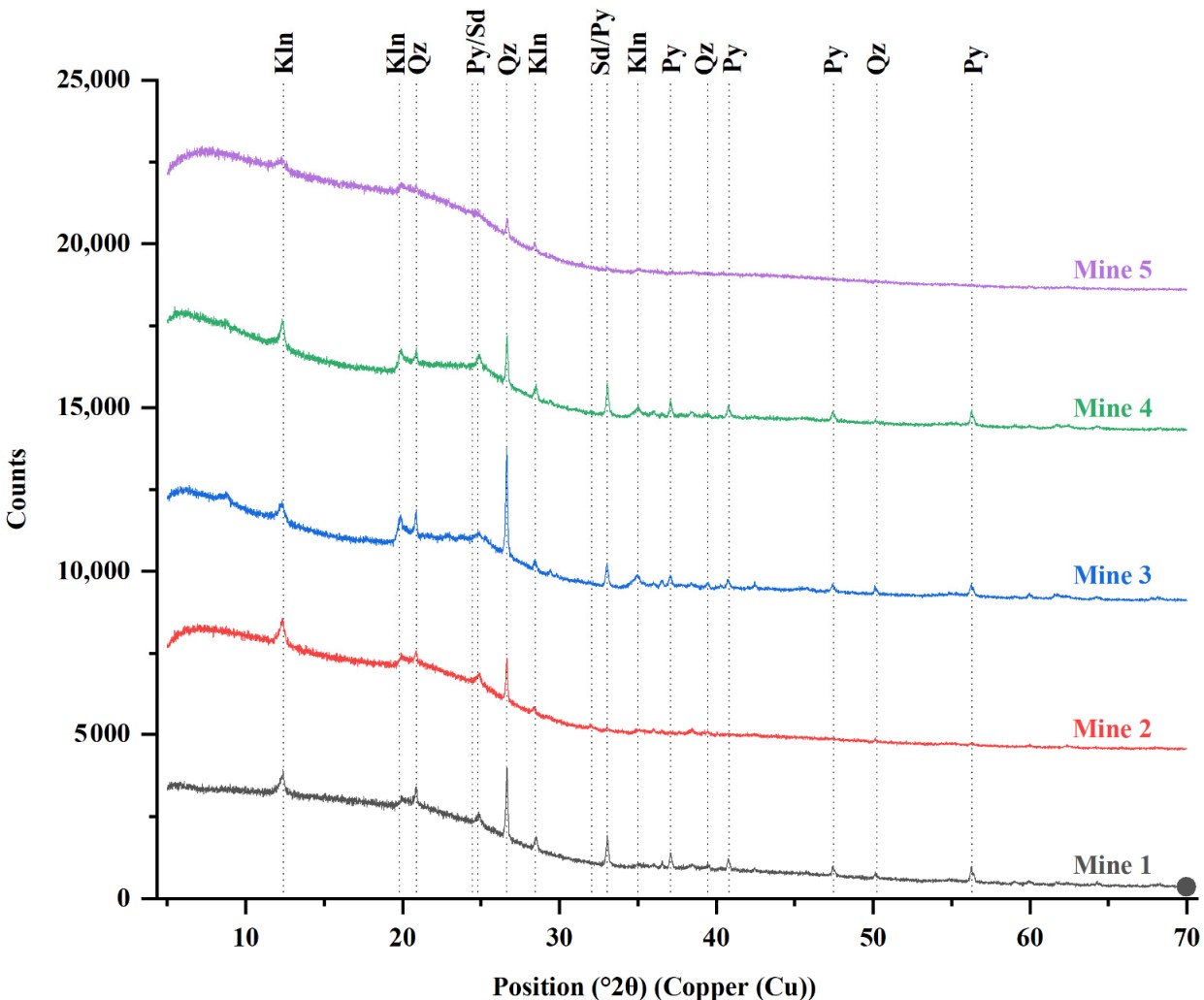

**Figure 2.** XRD patterns of coal samples from Mine 1, Mine 2, Mine 3, Mine 4, and Mine 5. Symbols according to [23,24]: quartz (Qz), kaolinite (Kln), siderite (Sd), pyrite (Py), and calcite (Cal).

**Table 3.** Minerals, compound names, and chemical formulas of the crystalline phases present in Mine 1, Mine 2, Mine 3, Mine 4, and Mine 5.

| | | | Mine ID | | | | |
|---|---|---|---|---|---|---|---|
| **Mineral** | **Compound Name** | **Chemical Formula** | **Mine 1** | **Mine 2** | **Mine 3** | **Mine 4** | **Mine 5** |
| Quartz | Silicon Oxide | $SiO_2$ | X | X | X | X | X |
| Kaolinite | Aluminum Silicate Hydroxide | $Al_2(Si_2O_5)(OH)_4$ | X | X | X | X | X |
| Pyrite | Iron Sulfide | $FeS_2$ | X | X | X | X | |
| Siderite | Iron Carbonate | $Fe(CO_3)$ | | X | | | |
| Calcite | Calcium Carbonate | $CaCO_3$ | | | X | | |

After plotting the raw data, a large wave in the spectrum at the beginning of the graph was observed. This hump in the XRD spectra is characteristic of amorphous materials, due to the random molecular structures [25,26]. It indicates the presence of considerable amounts of amorphous materials, corresponding to the carbonaceous matter.

The relative abundance of the mineral phases was obtained from the intensity displayed by each one in the XRD patterns. The same peaks from the different samples were compared to identify which mine had more content of each mineral. Some peaks in the spectra had influence from two minerals, so only the major peaks showing influence of single minerals were considered for comparison. This helps to have an idea of the relative abundance of the minerals among the mines, but not a measured value. The overall idea is that, comparing same peaks, the higher the intensity the higher the abundance of the mineral in the samples, since the beam found more atoms aligned to this direction. Table S2 shows the values of the counts for each mineral extracted from the software. In Figure 3, the counts per mineral phase were plotted for the five mines. The major peak was located at the position 2θ~26.65°, which corresponds to quartz. It is clear from Figure 2 that the peak for Mine 3 is significantly higher than for the rest of the samples, suggesting that Mine 3 has a higher amount of quartz in its composition. In the same way, Mine 5 has the smaller peak at this position, indicating lower amount of quartz in the sample.

Regarding kaolinite, from Figure 2, we can observe that the peaks for kaolinite (position 2θ ~ 12.38°) look to have similar intensities. This can be observed as well in Figure 3, where the major peaks for each mineral were compared. The relative amount within the mines is very similar, except for Mine 5, that had less counts in the diffraction pattern. Pyrite was not observed in Mine 5 and showed a low intensity in Mine 2. The other three mines showed similar contents of pyrite.

Mine 5 was a relatively cleaner sample, since it has only quartz and kaolinite and peaks can be barely seen in the diffraction pattern, indicating small amounts of minerals. With respect to the geographic locations, quartz and pyrite vary within the mines in the Appalachian region with no trends. The same observation was achieved for samples from the Rocky Mountains. Considerable differences were found within the Appalachian region mines since Mine 5 was collected in north–central Appalachian and Mine 3 and 4 were collected from northern Appalachia, which may indicate geological differences.

In underground mines, RCMD has contributions from different sources, mainly from the coal seam, host rock, diesel equipment, and rock dust. The final composition of RCMD is a mixture of all these components, providing in some cases, higher concentrations of quartz and other minerals (compared to the dust from only the coal seam), especially in the production areas [11]. In this study, only the coal samples from the seam were used, so small mineral contributions were expected in the dust samples.

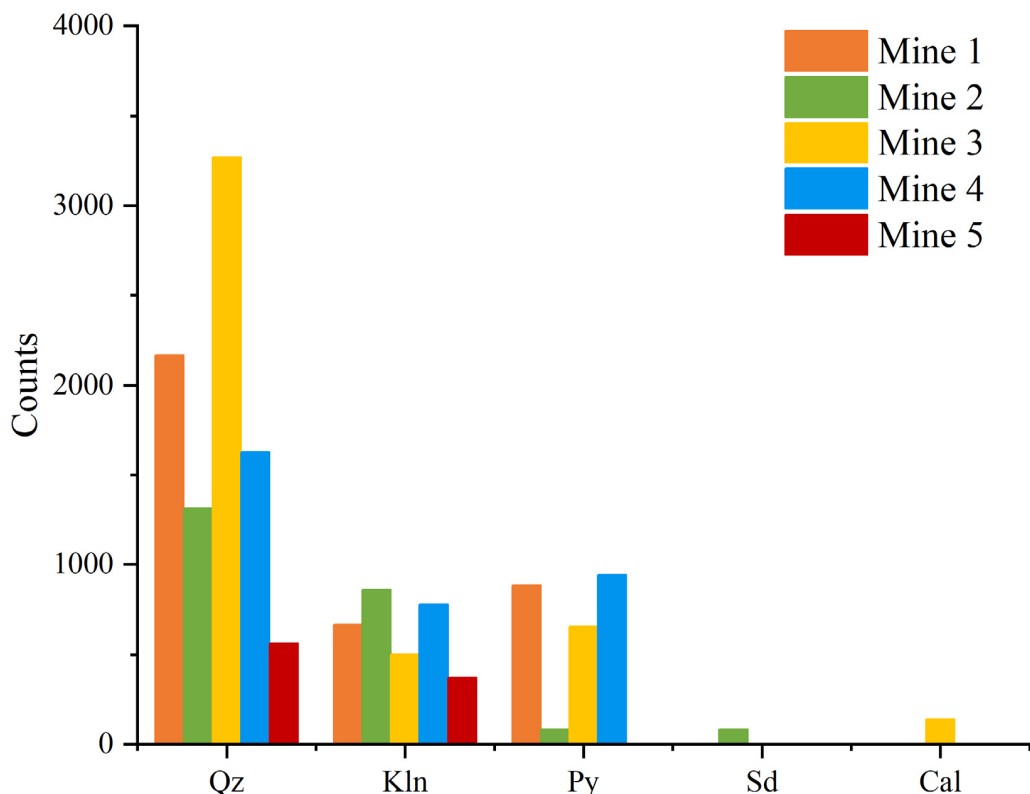

**Figure 3.** Relative abundance of the minerals in Mine 1, Mine 2, Mine 3, Mine 4, and Mine 5. Quartz (Qz), Kaolinite (Kln), Siderite (Sd), Pyrite (Py), and Calcite (Cal).

### 3.3. Elemental Composition

From the original raw data, Be, Co, Se, Mo, Ag, Cd, Sn, Sb, Tl, Th, and U were not included in the results since all of them showed readings under 10 ppb, which is lower than the limit of detection for the method with the dilution used. Additionally, after mass normalization, almost all of above-mentioned elements resulted in under 5 ppm, suggesting that these elements were in negligible concentrations or not present in the samples.

The SRM sample was digested completely, and the total concentration of the analyzed elements was obtained. The results from the method and the information of the SRM were compared and reliable data were obtained for 14 elements: Al, Si, Fe, Ti, Sr, Ba, Pb, Mn, Ni, Cu, As, V, and Cr (Table 4). Most of them met the allowable error described in the SRM information, when available, or remained between 1% and 15% when the error was not reported in the SRM data. Only Si, Al, and Fe were out of range, but still within 1%–15% of relative difference with the SRM, so they were included in the results.

**Table 4.** Summary of element content of dust samples.

| Mine ID | Al | Si | Fe | Ti | Sr | Ba | Pb | Mn | Ni | Cu | Zn | As | V | Cr |
|---|---|---|---|---|---|---|---|---|---|---|---|---|---|---|
| | % | % | % | µg/g | µg/g | µg/g | µg/g | µg/g | µg/g | µg/g | µg/g | µg/g | µg/g | µg/g |
| Mine 1 | 0.66 | 1.18 | 0.65 | 608 | 127.4 | 53.3 | 20.4 | 28.7 | 7.5 | 20.8 | 24.0 | 6.8 | 9.3 | 8.0 |
| Mine 2 | 0.73 | 1.13 | 0.20 | 503 | 21.6 | 8.9 | 23.5 | 7.2 | 7.4 | 11.7 | 12.6 | 1.0 | 4.8 | 7.1 |
| Mine 3 | 0.90 | 3.53 | 0.89 | 1114 | 41.2 | 56.6 | 16.6 | 24.4 | 21.8 | 20.4 | 52.7 | 27.8 | 36.9 | 26.3 |
| Mine 4 | 1.03 | 2.46 | 1.13 | 1040 | 69.2 | 45.0 | 8.7 | 8.0 | 16.0 | 13.0 | 36.3 | 14.8 | 55.4 | 26.1 |
| Mine 5 | 0.65 | 0.91 | 0.12 | 310 | 75.5 | 45.1 | 10.6 | 4.9 | 11.3 | 23.2 | 3.7 | 1.2 | 11.6 | 7.5 |

Li, Mg, K, and Ca showed relatively high concentrations but were not included in the results due to when using the SRM to verify the accuracy it was found that the relative difference was between 30% and 44%, very high to be accepted. These elements were under

or very close to their detection limit, so it may have led to spreading in the results. The information and results of the SRM are shown in Table S3. Zn and Sr were not reported in the SRM information, so it was not possible to verify their reliability, but the results were in the same order than the digested SRM in the lab, so they were included for reference. The results are shown in Figure 4 and are separated into major and trace element components and in Figure S1 at the same scale.

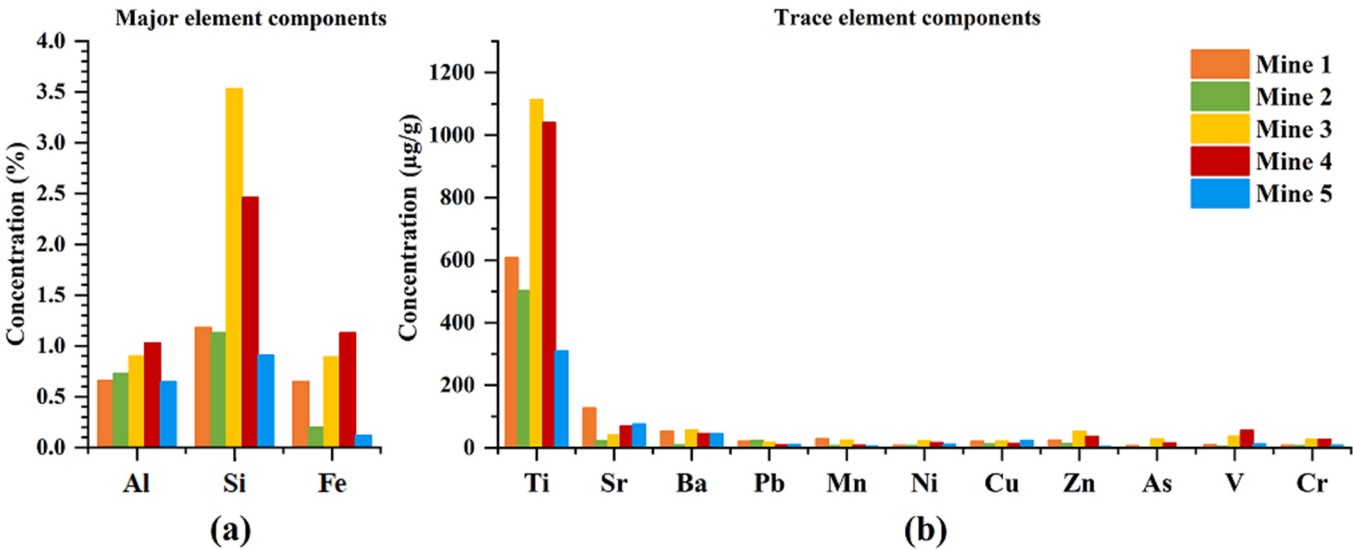

**Figure 4.** (**a**) Major and trace element components in the samples. (**b**) Trace element components in the samples.

Most of the elements had higher concentrations in Mine 3 and Mine 4, from the Appalachian region. Mine 5, overall, had the lowest concentrations, except for Cu and Sr, which were higher than in other mines. Al, Si, and Fe were the major elements in all the samples, corresponding to the major components of the minerals found in the XRD analysis. The rest of the elements were in very low concentrations, so it could be said that they are present in trace levels. Al showed similar concentrations among the coal from the different mines, similar to the XRD results. Fe was in a lower concentration in Mine 2 and Mine 5, and Si significantly higher in Mine 3, both in accordance with the XRD results as well.

*3.4. Surface Composition*

A typical coal dust sample exhibited strong peaks due to the carbon and oxygen with small peaks due to the iron also observed in the elemental content and dissolution studies. The C1s region was fitted using the asymmetric Doniach–Sunjic–Shirley profile as described in recent literature [21]. The C1s region shown in Figure 5a was dominated by a graphitic carbon peak as well as a sp$^3$ hybridized carbon peak. Carbon–oxygen bonds (C-O and C=O) corresponded to about 2.4% of the total carbon. This was much less than the total oxygen content obtained from the O1s/C1s ratio shown in Table 5. The O1s region was broad and did not allow an exact spectrum fitting, this suggests that coal dust samples also contained a significant amount of oxygen associated with hydroxyl groups or trapped water. The total oxygen content was comparable in all samples but for Mine 3 where it was markedly lower.

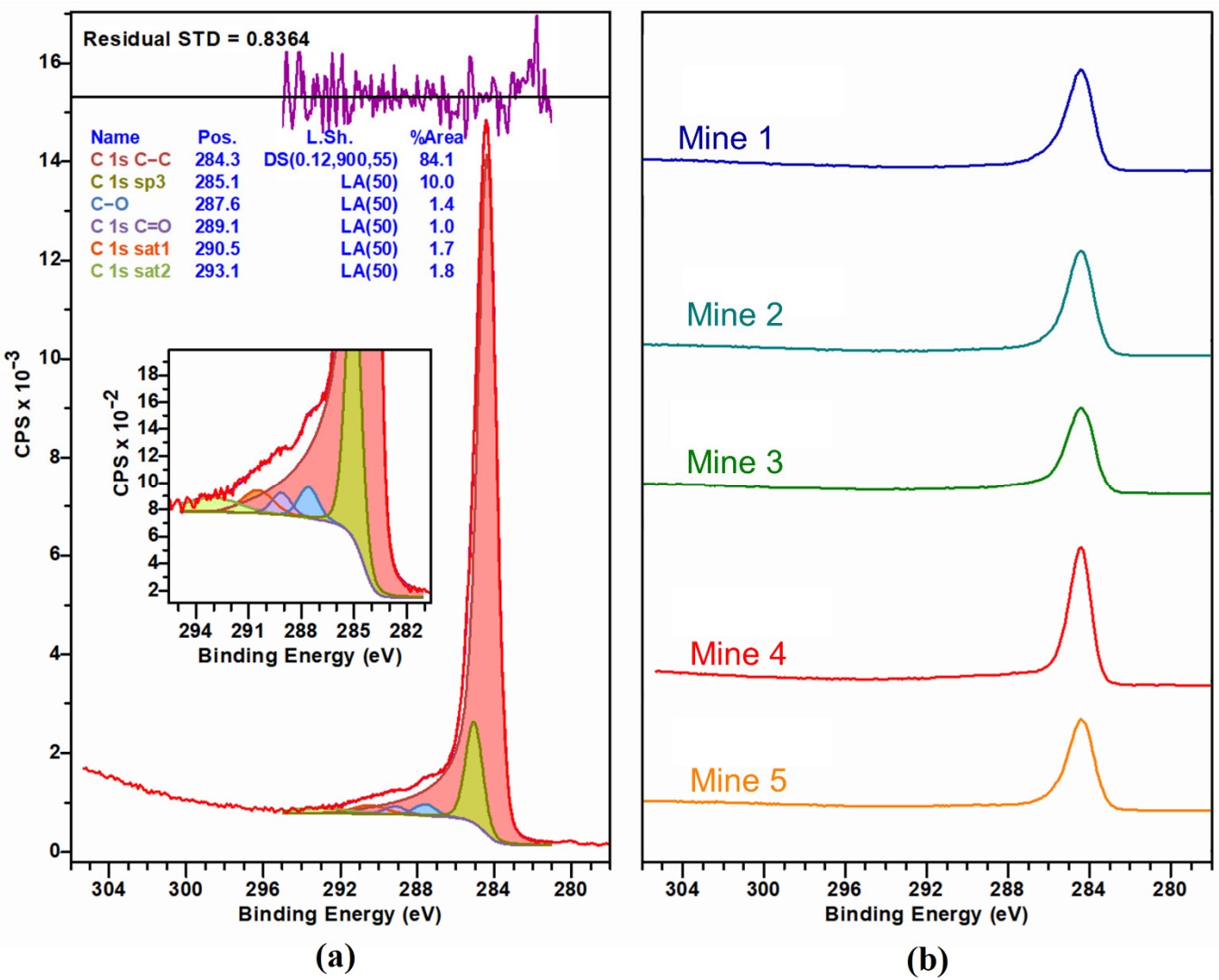

**Figure 5.** (**a**) High-resolution C1s spectrum of Mine 4, peak fitted with synthetic components based on the literature data [21]. (**b**) C1s spectra of the dust samples.

**Table 5.** Elemental composition on the surface of the coal dust.

| Mine ID | Al (2p) % | C (1s) % | O (1s) % | Si (2p) % |
|---------|-----------|----------|----------|-----------|
| Mine 1 | 0.90767 | 86.5527 | 11.7515 | 0.788068 |
| Mine 2 | 1.3677 | 87.0865 | 10.4115 | 1.13433 |
| Mine 3 | 0.827133 | 90.9843 | 6.74449 | 1.44407 |
| Mine 4 | 1.89306 | 87.7379 | 9.26774 | 1.10128 |
| Mine 5 | 0.174653 | 90.8217 | 8.27811 | 0.725521 |

The results are shown in Table 5. As expected, the main component in the samples surface was C1s. Si2p had a composition according to the results from the total digestion. In the case of Al, the results were not according to the total digestion. Mine 3 had a high concentration of Al in the total digestion, close to the concentration of Mine 4, but here the percentage of Al in Mine 3 was significantly lower. The same was seen for Mine 5, which had a similar concentration to Mine 1, but here it was shown to be around five times less than Mine 1. This may be explained with the minerals present in the selected samples. Al is related to the kaolinite, which occurs in nature packed in layers, so Al may not be exposed

in the surface in all the cases. Fe was found with the total digestion and XRD, but it did not appear exposed in the particles surface according to the XPS results.

When analyzing the trends by geographic location, it was not a clear pattern followed by the samples regarding Al and Si. The only significant differentiation was for O, which presented a lower exposure in the particle surface for the mines located in the Appalachian region.

### 3.5. Specific Surface Area and Micro-Pore Analysis

The specific surface area of dust samples was determined using a seven-points N2 adsorption isotherm and the measured surface area was found to be similar for all the samples, ranging from 6.80 to 7.77 $m^2/g$. The surface area was $7.77 \pm 0.45$, $7.66 \pm 0.15$, $6.80 \pm 0.10$, $7.11 \pm 0.23$, and $7.58 \pm 0.33$ $m^2/g$ for Mine 1, Mine 2, Mine 3, Mine 4, and Mine 5, respectively. Further, micro-pore analysis was performed, and all the samples were found to be mesoporous with a half pore size distribution found at 58.81 Å. The summary of the results is shown in Table 6.

**Table 6.** Specific surface area and half-pore width of samples.

| Mine ID | Specific Surface Area ($m^2/g$) | Half Pore Width (Å) |
|---|---|---|
| Mine 1 | $7.77 \pm 0.45$ | 58.81 |
| Mine 2 | $7.66 \pm 0.15$ | 58.81 |
| Mine 3 | $6.80 \pm 0.10$ | 58.81 |
| Mine 4 | $7.11 \pm 0.23$ | 58.81 |
| Mine 5 | $7.58 \pm 0.33$ | 58.81 |

The specific surface area and micro-pore analysis of the dust samples are important because they give the area that is in contact with the SLF in the dissolution experiments. The formation of the surface complexes is highly dependent on the exposed surface area and thus governs the dissolution process. The same mass (20 mg) was used for each dissolution experiment, therefore, a big difference in the specific surface area among the samples may influence the final dissolutions of the elements because it is presumed that a larger exposed area to the SLF can result in more dissolution, and finally biased results. In this case, the specific surface area of the samples was very close to each other, representing the minimum influence of the exposed area in the dissolution experiments.

### 3.6. Initial Functional Groups

FTIR spectra of coal dust were collected from a 4000 to 600 $cm^{-1}$ wavenumber range. In general, FTIR spectra of coal show four bands: 3800–3000 $cm^{-1}$ for the hydroxyl structures, 3000–2800 $cm^{-1}$ for the aliphatic structures, 1800–1000 $cm^{-1}$ for the oxygen-containing functional groups, and 900–700 $cm^{-1}$ for the aromatic structures. However, the spectra we collected showed no prominent peak in the hydroxyl and aliphatic region. Hence, spectra are reported within the range of 2000 to 700 $cm^{-1}$ wavenumbers. The obtained results are shown later in Figure 8 and their respective peak assignments in Table 9.

The absorption peak starting around 1200–1000 $cm^{-1}$ is the most obvious, which indicates that the presence of oxygen-containing functional groups in the coal dust is the highest. The peak coming at 1120–1080 $cm^{-1}$ is associated with S=O stretching. The peaks coming in the range of 1060–1020 $cm^{-1}$ are associated with Si-O-Si or Si-O-C stretching. The doublet coming at 799–779 $cm^{-1}$ is a representative peak for Si-O-Si bridging, which is the characteristic peak for low-temperature quartz and is also used for the quantitative determination of silica. The peak at 913 $cm^{-1}$ is associated with kaolinite, a common clay mineral. The peak at 1602 $cm^{-1}$ is for benzene ring C=C stretching, whereas the peak at 1445 $cm^{-1}$ is associated with antisymmetric –$CH_3$ deformation [27].

### 3.7. Dissolution in Simulated Lung Fluids

Variable dissolutions of elements across the samples and within the same SLF were found. This may be influenced by several factors such as the particle size, the surface area, the availability of the elements (elemental composition), the surface composition, and/or the mineralogy. Figure 6 shows the elements dissolving in GS and Figure 7 in ALF. Dissolutions were mass normalized, and each panel used the same scale to visualize the mines that presented more dissolution of the main elements. The data were fitted to the Langmuir-type model. Surface area normalization was carried out as well to eliminate the particle size and surface area influence and the results are shown in Figures S2 and S3 in the supporting information. The surface area was very similar for all the samples, so the same behavior of dissolutions was obtained in the results.

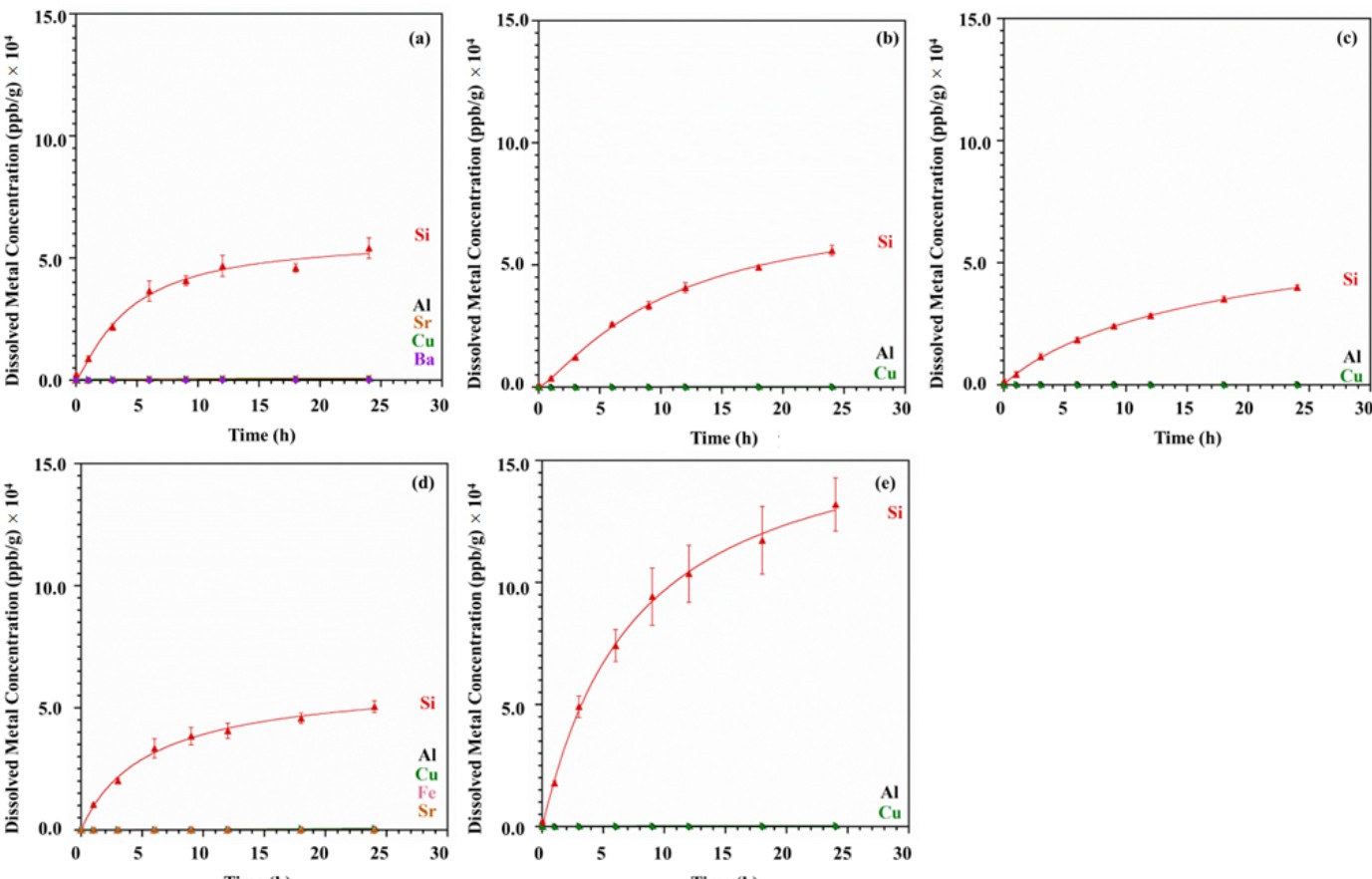

**Figure 6.** Mass normalized dissolution of metals as a function of time in GS from (**a**) Mine 1, (**b**) Mine 2, (**c**) Mine 3, (**d**) Mine 4, and (**e**) Mine 5.

The ICP-MS analysis showed Si being the most dissolved element in GS for all the coal samples. Significantly lower dissolutions of Al, Cu, and in some cases Fe, Ba, and Sr were found after 24 h of the dissolution experiment. For ALF, the higher dissolutions were found for Fe, Al, Si, and Cu, which showed increases in the concentration with the reaction time for all the coal samples. Very low dissolutions of Sr, Cr, Ba, Pb, and Ni were consistently found as well in ALF. The total dissolutions of the most dissolved elements after 24 h are shown in Table 7.

It was observed that some of the element dissolutions were influenced by the pH of the SLF used. Al, Fe, Sr, and Pb in general showed higher dissolutions in ALF than in GS, indicating the high influence of the pH of the SLF used. Al and Fe showed 4 to 16-folds and >50-folds higher dissolutions in ALF, respectively. Higher dissolutions for Si (9 to 48-folds higher) and slightly higher for Cu were found in GS. The higher dissolution of Si in GS may

be explained by the fact that part of the Si may come from the quartz ($SiO_2$) present in the sample. Quartz is an acid oxide, so it dissolves slowly in alkaline solutions [28]. The GS pH is 7.3, placed in the alkaline side of the pH scale. The ALF pH is 4.5, representing a more acid media for the Si dissolutions (giving less dissolution), but providing better conditions of pH for the dissolution of other components (i.e., Fe, Al, Sr, and Pb).

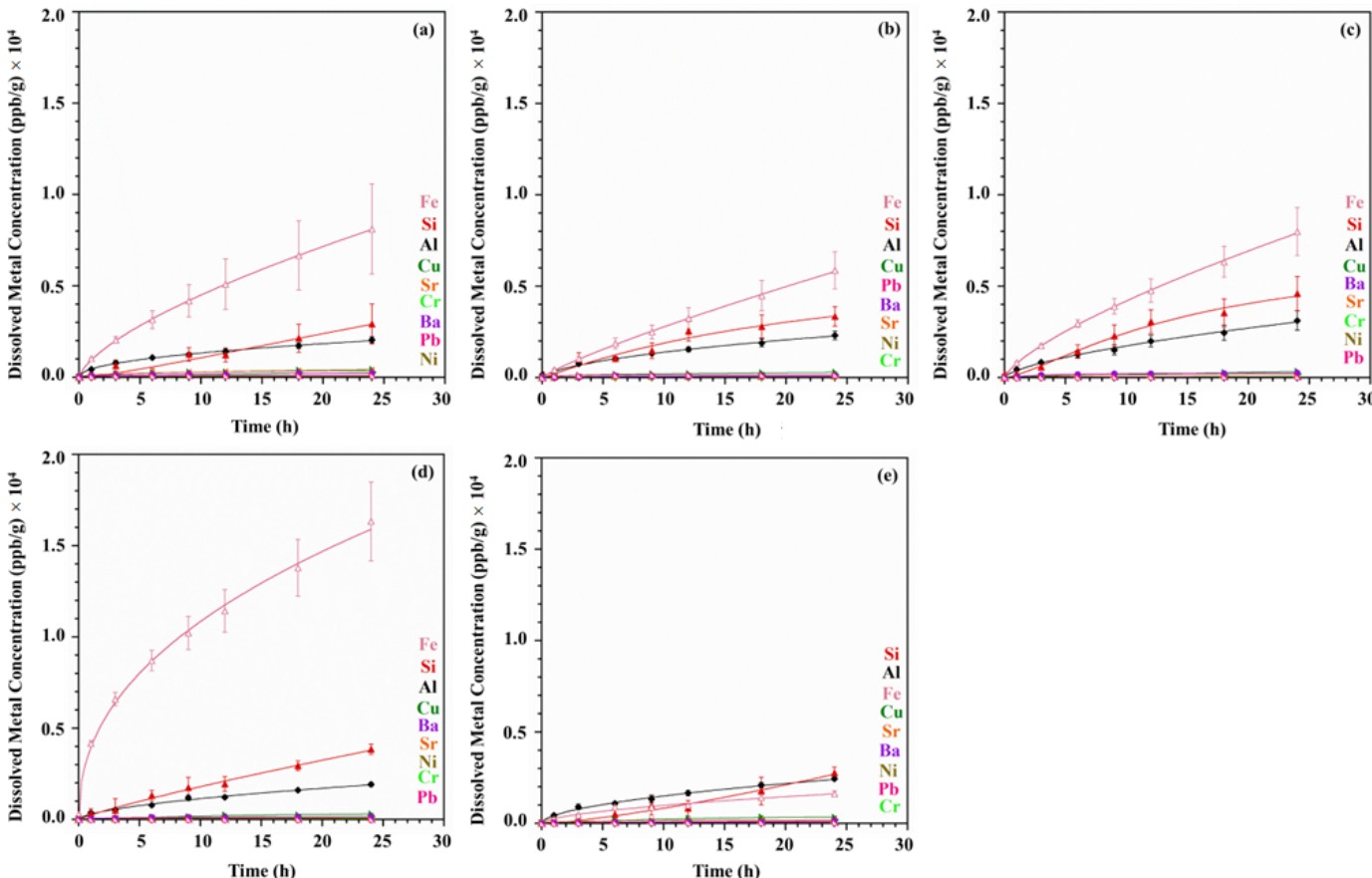

**Figure 7.** Mass normalized dissolution of metals as a function of time in ALF from (**a**) Mine 1, (**b**) Mine 2, (**c**) Mine 3, (**d**) Mine 4, and (**e**) Mine 5.

**Table 7.** Final metals concentration after 24 h of dissolution in SLF. Values in ppb/g.

| Mine ID | Al ALF | Al GS | Si ALF | Si GS | Fe ALF | Fe GS | Cu ALF | Cu GS | Sr ALF | Sr GS | Pb ALF | Pb GS |
|---|---|---|---|---|---|---|---|---|---|---|---|---|
| Mine 1 | 2316.8 | 142.6 | 3351.4 | 56,005.8 | 5879.2 | - | 291.5 | 323.6 | 83.1 | - | 160.2 | - |
| Mine 2 | 2058.8 | 555.8 | 2918.8 | 54,117.4 | 8117.8 | 7.7 | 435.9 | 692.4 | 391.8 | - | 109.9 | - |
| Mine 3 | 3135.5 | 210.2 | 4591.1 | 39,917.9 | 7993.1 | - | 354.4 | 245.0 | 188.9 | - | 47.6 | - |
| Mine 4 | 1925.6 | 468.1 | 3850.9 | 50,576.9 | 16,329.1 | 315.4 | 309.1 | 752.5 | 108.4 | - | 22.5 | - |
| Mine 5 | 2444.2 | 162.5 | 2764.3 | 131,965.8 | 1633.2 | - | 334.8 | 546.6 | 202.9 | - | 56.4 | - |

Characterization experiments indicated that Al, Si, and Fe were the main components of the samples, which gave the higher dissolutions as well in SLF, so overall, the initial availability of the metals in the samples plays an important role in the dissolutions in SLF (the more the availability, the higher the concentration after dissolution in SLF). However, comparing the magnitude of the initial concentration vs the total dissolution in SLF (availability normalization) for the major element components (Al, Si, and Fe) and some of the trace elements (Sr and Pb), a very small percentage of the initial amount was dissolved, representing less than 0.5% in most of the cases for both GS and ALF, many even below

0.1%. The availability comparison is shown in Table S4 in the supporting information. On the other hand, Cu, which was found as a trace element, having a lower availability in the samples, and, thus, a low concentration after the dissolutions in SLF, showed values ranging from 1.2 to 5.9% of the initial concentrations. So, Cu was more soluble than the other elements, since a higher percentage of the initial available Cu was digested in both GS and ALF. Sarver et al. [11] found much higher dissolutions of elements when RCMD samples collected in filter were dissolved in SLF. Si, Al, Cu, and Fe were found having 75, 26, 90, and 5% or less, respectively, of the total initial concentration dissolved in SLF. This previous study is based on dust samples collected on the filter, while the current research was conducted using bulk coal samples collected from the mines, and then prepared in the lab; therefore, the composition of the dust is different. According to the mean mineralogy distributions found for the samples collected in the filter (larger particles: 400–10,000 nm), the carbonaceous mineralogy class represented less than 15% of the composition for all the mines and sample locations studied [11]. Characterization results indicate that samples used in this study were mainly carbonaceous material with very few mineral contributions. In the case of the major elements found by Sarver et al. [11], the total element concentrations (acid-soluble) ranged from 10 to 200 mg/g, while the samples used in this study ranged from 1.2 to 35.3 mg/g (0.12%–3.53%) as shown in Table 4. These significant differences in sample compositions might explain the difference in the results between the two studies.

Dissolutions of Al in GS and Si, Fe, and Pb in ALF were higher when the initial availability was higher, indicating that dissolutions in these cases are directly related to the initial availability of the metal in the sample. This behavior was not observed in the rest of the dissolutions, which in general were led by the pH of the solution. Only for Cu it was observed that the dissolutions were not driven by the initial availability nor pH of the solutions. In this case, the dissolution may have been driven by the chemical affinity of the Cu with the free ions in the SLF solutions. The mode of occurrence and oxidation state of the elements in the RCMD have an important role as well in the dissolution process, improving or impairing their affinity with the free ions and solvent agents in the SLF solutions.

The Sarver et al. [11] study, associated the dissolution in SLF of these elements with the contribution of the geologic strata in the mine or the rock dusting products applied to the RCMD [11]. This study only studied coal from the seam (no strata or rock dust influence); thus, it can be concluded that the coal seam also contributes to the final concentrations of elements found after dissolution experiments in the SLF, but probably in small quantities.

When comparing by geographic location, a clear trend was not observed, since the dissolutions were led by different factors (Table 8). Even if samples from the Appalachian region had a higher initial availability in most of the cases, the dissolutions experiments had a varied behavior in the final concentration after being dissolved in SLF.

**Table 8.** Factors influencing dissolutions in SLF. Av: availability.

| Al | | Si | | Fe | | Cu | | Sr | | Pb | |
|---|---|---|---|---|---|---|---|---|---|---|---|
| **ALF** | **GS** | **ALF** | **GS** | **ALF** | **GS** | **ALF** | **GS** | **ALF** | **GS** | **ALF** | **GS** |
| pH | pH + Av. | Av. | pH | Av. + pH | pH | Possible Affinity | Possible Affinity | pH | pH | pH | pH |

In summary, from the dissolution experiment, it was obtained that the main elements dissolving from the dust samples when in contact with the SLF were Al, Si, Fe, Cu, Sr, and Pb. The overloading in the human body of Al, Fe, Cu, and Pb have been related to different diseases and health issues as described below.

In humans, $Mg^{2+}$ and $Fe^{3+}$ are replaced by $Al^{3+}$, which causes many disturbances associated with intercellular communication, cellular growth, and secretory functions [29,30]. Al has been found to be very harmful to nervous, osseous, and hemopoietic cells [29]. Al also accumulates in the kidney, producing renal function [31–33]. Exposure to quartz as RCS (present in the samples) has been related to the development of chronic renal disease [34].

Fe and Cu are essential elements for humans. Fe plays an important role for fundamental vital activities (such as growth and survival) [29,35–42], and Cu in the function and maintenance of the human immune system [43–47]. While Cu and Fe are an essential nutrient for humans, they can pose risks to human health with elevated exposure [35,39,46,48]. An excess of Cu and Fe aggravates oxidative stress, which leads to accelerated tissue degeneration and DNA damage [35,47,49]. Fe overload may be harmful, causing carcinogenesis. Fe toxicity is largely based on its ability to catalyze the generation of radicals, which attack and damage cellular macromolecules and promote cell death and tissue injury [35,39,40]. Hemochromatosis, hepatocellular cancer, iron-loading anemias, dietary iron overload, and chronic liver disease are conditions related to Fe overload [35,40]. Furthermore, Fe dysregulation is closely associated with the initiation and development of several malignant tumors, including lung cancer [39,41,42].

An excess of free Cu ions can cause damage to cellular components and reduced cell proliferation [47,49]. In the most severe forms, Cu toxicity leads to rhabdomyolysis, cardiac and renal failure, methemoglobinemia, intravascular hemolysis, hepatic necrosis, encephalopathy, and ultimately death [49].

Al, Fe, and Cu are related to neurodegenerative changes [35,44,50]. Al and Fe have been linked to neurological disorders including Alzheimer and Parkinson's disease [30–32,35,50–53]. Cu toxic levels in the brain have been reported to cause apoptosis, astrocytosis, impaired learning and memory, cognitive dysfunction, and accelerate disease progression [46].

Pb is the second most toxic metal after Arsenic (As) and is considered carcinogenic (Group 2B) to humans [54–56]. In adults, Pb causes cardiovascular, central nervous system, kidney, and fertility problems [55,57]. Long-term exposure to Pb is associated with immune dysfunction and may affect kidneys, heart, liver, brain, and lung [58,59]. Pb exposure has been shown to induce oxidative stress and the altered expression of genes related to inflammation. Previous studies have reported increased incidences of lung cancer among workers exposed to Pb [59–63].

Si and Sr have not been reported as harmful for humans [64,65]. Sr can replace Ca in bones and in teeth when entering the bloodstream [66], but its accumulation can be slowly eliminated from the body, taking long periods [65]. Silica is used widely in the food and beverage industry as a food additive, so the human body obtains large loads of Si from dietary sources. Relatively insoluble forms of silica can release small but meaningful quantities of silicon into biological compartments. Still, Si exposure to humans is limited and largely in chemical forms that are not readily absorbed nor bioavailable [67]. Opposite to toxic, silicic acid has been associated to beneficial effects for bone [68], and also to the reduction of Al toxicity and risk of Alzheimer's disease, due to its high affinity to Al [69,70].

The quantities of metals dissolved in SLF that were found are lower than the recommended dietary allowance (RDA), tolerable upper intake levels (UL), or average intake (AI) usually found for these elements, which are in almost all the cases in the order of mg/day (Al-AI: 10 mg/day [71], Fe-UL: 1.1 mg/day [64], Cu-UL: 10 mg/day [64], Sr-AI: 1.9 mg/day [72], and Pb-UL: 1.75 mg/week [73]), except for Si, for which there is no evidence of adverse health effects or UL [64].

When compared with the results, the values found were lower since the results are given in ppb/g. In addition, the concentrations obtained are per gram of coal dust, so considering that the permissible exposure limit (PEL) for RCMD is 1 mg/m$^3$ [1], if we set a scenario where a mine folds five times the PEL (i.e., 5 mg/m$^3$), it would take a long time for a miner to inhale 1 g of RCMD. An average breath carries around 0.5 L of air [74], and the normal number of respirations for a healthy adult ranges from 12 to 20 breaths per min [75]. This means that, considering the extreme (20 breaths/min), a person could breathe around ~5 m$^3$ of air in 8 h (normal shift), resulting in a total dose of ~25 mg/8-h of RCMD, so it would take several days to achieve 1 g inhaled, not considering the fraction that is exhaled or expelled from the body. Thus, the concentrations found in the dissolution experiments are difficult to reach in short periods from RCMD, and more, concentrations in toxic levels.

However, the direct release of these elements to the cells and the direct contact with tissues may play a role in the cell damage. It was found in the literature that free Cu ions can cause cell damage, replacements by $Al^{+3}$ cause disturbances in the cell communication and growth, and Pb exposures altered the expression of genes related to inflammation, so the effects of the direct contact of these elements with the cells should be further studied to determine how they may affect their functionality and be involved in the development of diseases.

### 3.8. Changes in Functional Groups

FT-IR analysis was carried out on the samples after dissolution in SLF experiments. The spectra collected for the coal dust that has been dissolved in SLF and recollected showed almost similar spectral bands to the initial spectra found in the samples before dissolution, but with lower intensities. The spectra of the dust after dissolution indicate oxygen-containing functional groups are the most consumed. The spectra obtained for samples before and after dissolution in SLF are showed in Figure 8 and the peak assignments in Table 9.

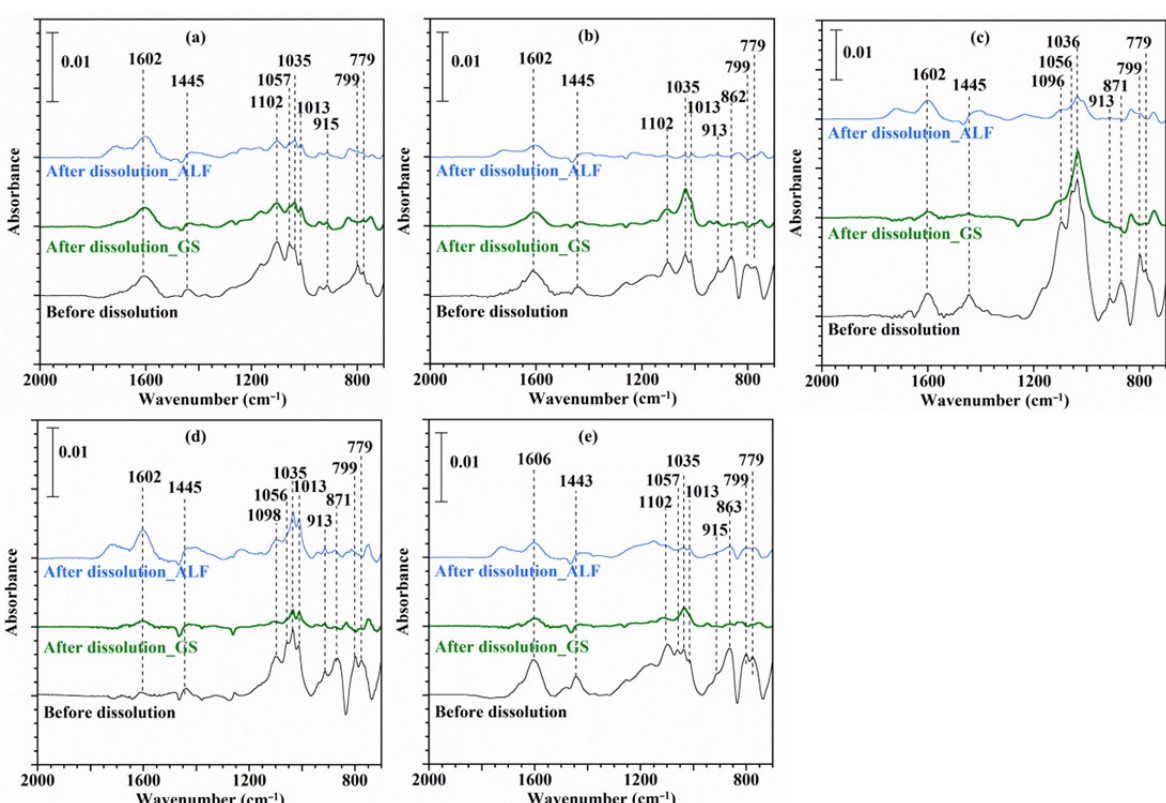

**Figure 8.** FTIR spectra of (**a**) Mine 1, (**b**) Mine 2, (**c**) Mine 3, (**d**) Mine 4, and (**e**) Mine 5.

**Table 9.** Peak assignments from FT-IR.

| Wavenumber (cm$^{-1}$) | Peak Assignment |
| --- | --- |
| 779 and 799 | Quartz |
| 913 | Kaolinite |
| 1000–1200 | Si=O, Si-O-Si, Si-O-C, C-O-C |
| 1445 | Antisymmetric–CH$_3$ deformation |
| 1602 | Benzene C=C stretching |

*3.9. In Vitro Inflammatory Response*

Existing literature has focused on the development of pulmonary fibrosis in pneumoconiosis, largely examining the role of fibroblasts in the development of CWP [76–78]. Instead of merely examining the role of fibroblasts, we assessed the impact of coal-based PM10 on epithelial (A549), neutrophilic (HL-60), and macrophage (THP-1) cells. The results of the PM10 coal dust in vitro exposures for cells HL-60, A549, and THP-1 are shown in Figure 9, Figure 10, and Figure 11, respectively.

After 4 h of in vitro dust exposure, HL-60 cells indicated mostly a decrease in cytokine expression. Interferon gamma (IFN-γ), interleukin-10 (IL-10), interleukin 12p70 (IL-12p70), interleukin-13 (IL-13), interleukin-4 (IL-4), and interleukin-6 (IL-6) demonstrated a diminished expression across low (5 μg/mL), medium (10 μg/mL), and high (20 μg/mL) treatment groups, relative to controls (no dust exposure). These cytokines are biological signaling molecules frequently involved in chronic inflammatory conditions, including CWP. Overall, the three concentrations did not demonstrate a dose response, only IL-1β showed a decrease in the cytokine expression when the dose increased for almost all the mines. Interestingly, IL-2 expression mostly increased across treatment groups, with Mine 2 and Mine 5 demonstrating the greatest relative IL-2 expression. IL-8 and IL-1β indicated mixed results, with both up and downregulation across all three treatment groups.

For A549 cells, after 4 h of in vitro dust exposure, they showed an increase in cytokine expression for all the mines across both low (10 μg/mL) and high (100 μg/mL) treatment groups. IL-10 demonstrated a remarkably high increased expression when exposed to low concentrations, except for Mine 3. IL-8 showed the lowest response to the PM10 exposures, as well as IL-1β, which showed a very low upregulated expression for most of the mines, except for Mine 1 at a high concentration. IL-10, IL-12p70, IL-6, and TNF-α indicated a dose response, showing a decreased expression when the dose was increased. The rest of the cytokines showed a mixed dose response, with both increasing and decreasing expressions when treated with a higher concentration.

THP-1 cells also displayed an increase in cytokine expression for all the mines after 4 h of in vitro dust exposure and across both low (10 μg/mL) and high (100 μg/mL) treatment groups. Overall, the two concentrations did not demonstrate a dose response, only IL-1β showed a decrease in the cytokine expression when the dose increased.

Selected cytokines were selected based on their role in pneumoconiosis. Chronic inflammation is a key symptom of pneumoconiosis in lung tissues and bronchoalveolar lavage fluid [79]. Pneumoconiosis is characterized by pulmonary injury and the recruitment of inflammatory cells such as monocytes, macrophages, and neutrophils [80–82]. Neutrophils are known for being the most abundant leukocytes in the blood, and for that, they are used as the first line of defense in the immune system in several situations. From the circulation, they are quickly mobilized to sites of inflammation [83,84]. In pneumoconiosis, neutrophils respond to several inflammatory cell chemoattractants generated by activated macrophages. Neutrophils migrate from the vascular compartment to the alveolar space [80,81]. Once they are inside the alveolar space, recruited neutrophils secrete toxic oxygen radicals or proteolytic enzymes. As a result, they induce an inflammatory response by avoiding lung colonization from agents such as silica, asbestos fibers, and coal dust [80,81,85].

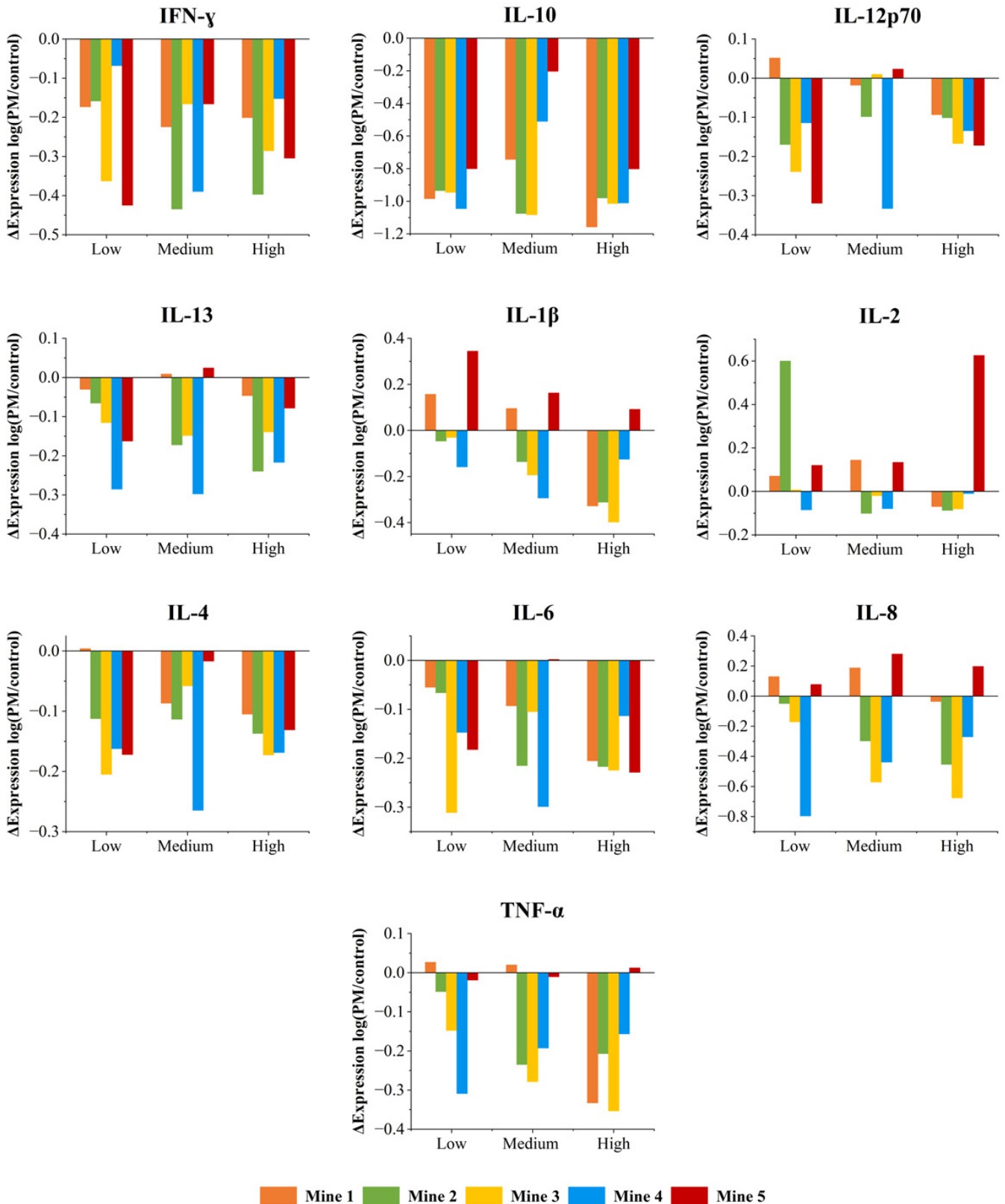

**Figure 9.** Results of PM10 coal dust exposures to HL-60 cells using low (5 μg/mL), medium (10 μg/mL), and high (20 μg/mL) concentrations.

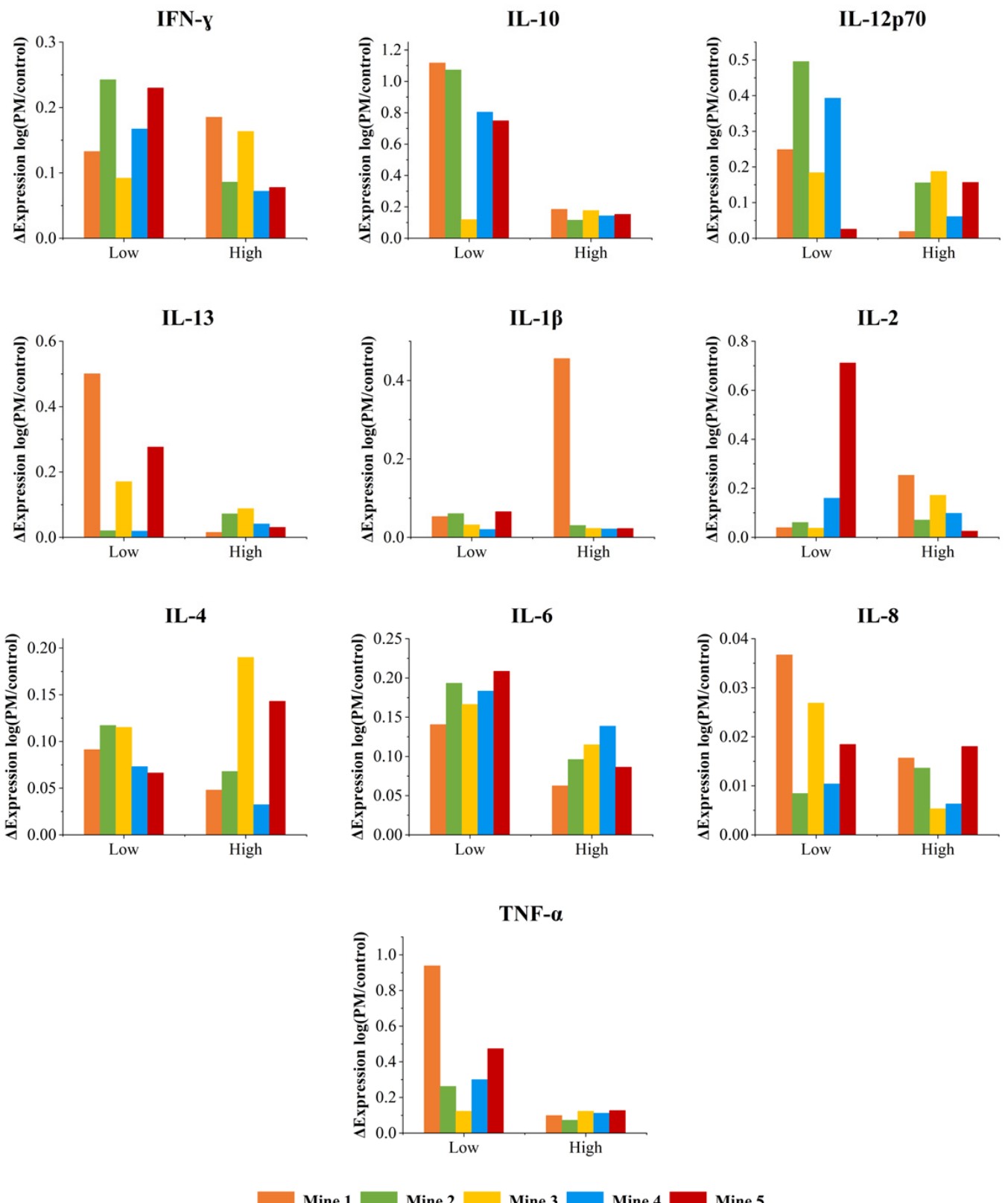

**Figure 10.** Results of PM10 coal dust exposures to A549 cells using low (10 µg/mL) and high (100 µg/mL) concentrations.

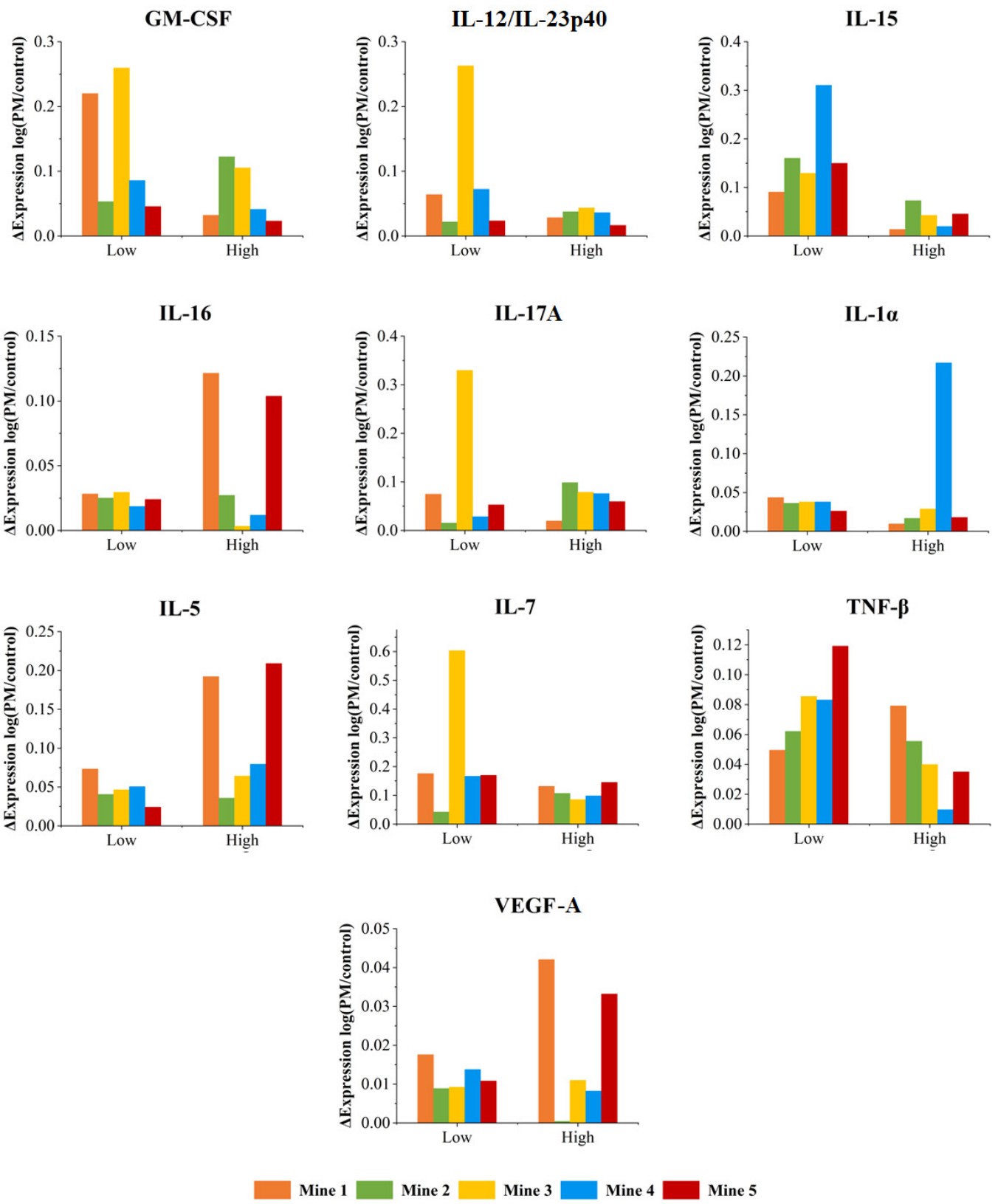

**Figure 11.** Results of PM10 coal dust exposures to THP-1 cells using low (10 µg/mL) and high (100 µg/mL) concentrations.

Dust particles under 5 µm escape mucociliary clearance and deposit in the terminal bronchioles and alveoli [85,86]. Then, lung epithelial cells play a huge role in avoiding

lung colonization by these agents (silica, asbestos fibers, or coal dust). They recognize microbial molecules through specialized receptors, such as toll-like receptors, pattern recognition receptors (PRRs), and CD14 receptors [87]. The detection of foreign material by the epithelial cells triggers the immune system response, resulting in the activation of alveolar macrophages, which accept the smaller particles via phagocytosis and collect in the interstitium along the perivascular and peribronchiolar regions of the lung [86]. It also results in the release of cytokines that stimulate inflammation, such as IL-1 and TNF-$\alpha$, the generation of free radicals, and the augmentation of cell-signaling pathways. As a result, the different cytokines result in the promotion of fibrosis [85,86,88]. Following this, once the inflammation process is complete, the fibrotic process initiates by stimulating growth factors. Then, type 1 pneumocytes grow over the collected alveolar macrophages and enclose them in the interstitium. Fibroblasts become stimulated to cause fibrosis and tissue remodeling by producing ECM and matrix metalloproteinases [85]. Fibrocytes can also induce chemotaxis and attract inflammatory factors and chemokines to increase the immune response trying to prevent the establishment of these pathogenic microorganisms [85]. By the end of the fibrotic process, there is an overproduction of fibronectin and collagen, resulting in scar tissue formation in the lungs [85], which is less compliant than normal lung tissue, making it harder to breathe over time [89]. These reactions take place for years before showing any symptoms [89]. Heppleston's theory states that the death and disintegration of macrophages engulfing dust results in fibrosis [79].

TNF, IL-6, and IL-8 are some of the mediators that may be involved in the pathogenesis of coal workers' pneumoconiosis [76]. TNF participates in the initiation and regulation of inflammatory reactions, as well in the regulation of fibrotic reactions, while IL-6 and IL-8 in the chemotaxis and activation of lymphocytes and the chemotaxis of inflammatory cells, respectively [76]. Previous literature has found TNF-$\alpha$, IL-6, and IL-1 highly related to CWP and silicosis after conducting in vitro, in vivo animals, and human studies from patients with these lung diseases [76,90–94]. The increase in both TNF-$\alpha$ and IL-12p40 was associated with silicosis development and severity [94]. TNF seem to play a crucial role in the control of the inflammatory and fibrotic response of the lungs to RCMD and RCS [76,95,96]. Furthermore, it has been found that IL-8 (important as neutrophil attractants and adhesion molecules), is associated with PMF [81].

Davis G.S. et al. studied IFN-$\gamma$ (a lymphocyte cytokine that acts in macrophage activation) and IL-4 (involved in the differentiation of T cells and eosinophilic inflammation [97]) in mice exposed to silica, finding an immune–inflammatory response from IFN-$\gamma$ but not from IL-4 [98]. IFN-$\gamma$ also may help in the production of fibroblast growth factors by macrophages, but its influence remains in discussion [76].

IL-12 has multiple biological activities, and it is a key factor that drives Th1 responses and IFN- production. IL-12 may provide protection against bacterial and viral infections. IL-13, which is linked to inflammatory diseases and appears to contribute to the development of pulmonary fibrosis and the formation of granulomas, was found upregulated in a study when mice were exposed to silica, relating this cytokine to lung diseases [99].

IL-2 is a signaling molecule involved in T-cell proliferation. Previous literature suggests that bronchoalveolar lavage fluid samples from pneumoconiosis patients have tested positive for IL-1$\alpha$ and $\beta$, TGF-$\beta$, IL-5, IL-2, and IL-10 [100]. IL-12 has a variety of biological functions and is a critical regulator of Th1 responses and IFN-$\gamma$ production; also, it may play a key role in protecting against bacterial and viral infections [101].

In these current results, TNF-$\alpha$, IL-2, IL-1$\beta$, IL-8, IL-6, and IL-10 were upregulated mainly for Mine 1 and Mine 5 in the neutrophilic cells, and for all the mines in epithelial cells, but showing a higher expression for Mine 1 and 5 as well in most of the cases. IL-1$\alpha$ and IL-5 in the macrophage cells were also upregulated, showing an especially high expression for Mine 4 in IL-1$\alpha$ and for Mine 1 and Mine 5 in IL-5 at high concentrations. IFN-$\gamma$ and IL-4 were upregulated in epithelial cells for all the mines but were downregulated in neutrophilic cells, not displaying any trend.

From the data obtained, in some of the cytokines studied in the neutrophilic cells, an immune suppressive effect was found (a downregulated response), but in general the cytokines of the epithelial, macrophage, and some of the neutrophilic cells showed an upregulated response, indicating an inflammatory response when exposed to low and high concentrations of PM10 coal dusts. This inflammatory response obtained from most of the cytokines may have led to the activation of the mechanisms mentioned that produce the scar tissue formation and, thus, lung diseases. Results also indicate that Mine 1 and Mine 5 may produce higher inflammatory stimulation since they were found consistently upregulated and/or with a higher expression in the cytokines that the literature links to inflammation and pneumoconiosis. Based on our results, further research is warranted on cytokine production in non-fibroblast cell types in pneumoconiosis models, including neutrophils, lung epithelial cells, and macrophages.

## 4. Conclusions

This study provides characterization and bio-accessibility information of RCMD from two regions in the US (the Appalachian region and the Rocky Mountains). This study analyzed the characteristics and toxicity of the RCMD fraction coming only from the coal seam and compared the differences of samples coming from the Appalachian region and the Rocky Mountains to identify the existence of trends related to the geographic location. From results obtained, the main conclusions are:

- The particle size distribution of the samples showed to be finer for those coming from the Appalachian region. Appalachian region samples were suggested to have more minerals and higher elemental concentrations, which would indicate that these samples would be more resistant to the reduction process, but otherwise as expected, these samples reached finer samples when reduced under the same procedure. For samples from the Appalachian region, even particles smaller than 1 micron were found, but not for samples from the Rocky Mountains;
- The XRD experiment showed that quartz, kaolinite, and pyrite were the main mineral components of the samples. These results were in accordance with the elemental content results, where Si, Al, and Fe were the elements with higher concentrations in the samples, and the main element components of the minerals observed in the XRD. Additionally, XRD and elemental content results showed in general that samples from the Appalachian region had more mineral and elemental contents compared to the samples from the Rocky Mountains;
- As for XRD analysis, Si followed the same trend for its exposure in the surface as the elemental content, but Al did not show any trend related to the initial concentration or the geographic location, indicating that not the same proportion of Al was exposed in the particle surfaces, and that Al atoms may be packed inside the particles or the characteristic layers of the clay minerals (kaolinite). Fe was not observed in the particle surfaces, but it was dissolved later in the dissolution experiment, so the Fe atoms, even if they were not exposed in the surface, were reached by the SLFs and partially digested;
- Kaolinite and quartz showed decreases in the peaks of the FT-IR spectra after dissolution in both GS and ALF, indicating that these minerals are actually being digested by the SLFs and the dissolution of Al and Si obtained in SLF came from them;
- The main factors influencing the dissolutions were the pH of the SLF and the initial availability of the elements in the samples. The specific surface area did not affect the general behavior of the dissolutions;
- The elements with the higher availability in the samples (AL, Si, and Fe) gave the higher total dissolutions in the SLFs, but the percentage dissolved from the initial contents did not exceed 0.5% in most of the cases. In contrast, Cu (trace elements) was dissolved from 1.2 to 5.9% of the initial content in the samples, so it can be concluded that, normalized to the initial availability, Cu is more bio-accessible than the other elements;

- The toxicity of the samples based on the metal dissolutions could not be related to the geographic location, since the factors influencing the dissolutions in SLF varied from the different samples and within the elements dissolved. So, the higher incidence of lung diseases in the Appalachian region may be related to other factors such as the exposure to RCMD, the particle size distribution of the actual RCMD in each mine, and the mineral contributions from the different sources in the mine to the RCMD that the miners inhale, which may be significantly different in each region;
- In vitro studies indicated a proinflammatory response of the cytokines studied, especially in the epithelial and macrophage cells, which suggests a possible participation from these cell types in pneumoconiosis and lung diseases development.

**Supplementary Materials:** The following supporting information can be downloaded at: https://www.mdpi.com/article/10.3390/min12070898/s1, Table S1: Composition of the SLFs used; Table S2: Percentages of the relative mineral abundance, total counts, and counts per mineral in XRD data; Table S3: SRM verification ($\mu$g/g); Table S4. Percentage dissolves from the initial availability of the elements (dissolved/available); Figure S1. Elemental content of the sample in same scale; Figure S2. Mass and surface area normalized dissolution of metals as a function of time in GS from (a) Mine 1, (b) Mine 2, (c) Mine 3, (d) Mine 4, and (e) Mine 5; Figure S3. Mass and surface area normalized dissolution of metals as a function of time in ALF from (a) Mine 1, (b) Mine 2, (c) Mine 3, (d) Mine 4, and (e) Mine 5.

**Author Contributions:** Conceptualization, V.S., M.D., P.R., G.R. and K.Z.; methodology, M.D. and V.S.; validation, V.S. and M.D.; formal analysis, V.S., M.D. and N.F.; investigation, V.S., M.D., Q.J., A.C., G.R., M.H., A.M. and J.B.; resources, M.H., M.R., K.Z., P.R. and G.R.; writing—original draft preparation, V.S.; writing—review and editing, V.S., M.D., G.R., P.R., K.Z. and M.R.; visualization, M.D., V.S. and G.R.; supervision, P.R. and G.R.; project administration, P.R.; funding acquisition, P.R., G.R., M.R. and K.Z. All authors have read and agreed to the published version of the manuscript.

**Funding:** This research was funded by National Institute for Occupational Safety and Health (NIOSH), grant numbers #75D30119C06390 and #75D30121C12182, and by National Institute of Environmental Health Sciences (NIH/NIEHS) under grants R21 ES032432 and R00 ES029104. The views, opinions and recommendations expressed here are solely those of the authors and do not necessarily reflect the views of NIOSH. Mention of trade names, commercial products or organizations does not imply endorsement by the authors nor the funding organization.

**Data Availability Statement:** Not applicable.

**Acknowledgments:** The authors thank Maria Pineda, and Carlha Barreto for helping with the sample preparation, and Virgil Lueth, Bonnie Frey, and Kelsey McNamara at the New Mexico Bureau of Geology for providing training to perform XRD and total microwave digestion experiments.

**Conflicts of Interest:** The authors declare no conflict of interest. The funders had no role in the design of the study; in the collection, analyses, or interpretation of data; in the writing of the manuscript; or in the decision to publish the results.

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
