# Peer review of "Characterization and Toxicity Analysis of Lab-Created Respirable Coal Mine Dust from the Appalachians and Rocky Mountains Regions"

_minerals, doi:10.3390/min12070898_

Round 1

Reviewer 1 Report

The article presents the results of the characterization and toxicity analysis of respirable coal mine dust (RCMD) in some representative mines in the USA. The idea presented is very interesting, and the results shall be implemented. Although the paper is very well-written, it needs a revision in some points as indicated below:

·       The bibliography presented is appropriate and exhaustive. However, authors shall review the citation in the text, for instance, lines 130, 280, 316, and so on.

·       The paper must be integrated with maps where the position of the mines investigated and the sampling points are clear.

I believe that overall, the paper deserves to be published in a prestigious journal like Minerals after the indicated improvements.

Author Response

We appreciate your time and effort in reviewing our paper. we carefully applied your comments and address them in the manuscript. 

  1. The bibliography presented is appropriate and exhaustive. However, authors shall review the citation in the text, for instance, lines 130, 280, 316, and so on.

Answer: Thank you for your comment. We addressed your comment in the manuscript.

  1. The paper must be integrated with maps where the position of the mines investigated and the sampling points are clear.

Answer: Thank you for your comment. Due to the non-disclosure agreement (NDA) agreement, we are not allowed to indicate the exact location of the mines. However, the states where the mines are located were added to Table 1 to give a better idea of the locations.

Reviewer 2 Report

I’ve finished a review of the paper minerals-1806727 titled Characterization and Toxicity Analysis of Lab Created Respirable Coal Mine Dust from the Appalachians and Rocky Mountains Regions written by the authors Vanessa Salinas , Milton Das , Quiteria Jacquez , Alexandra Camacho , Katherine Zychowski , Mark Hovingh , Alexander Medina , Gayan Rubasinghege , Mohammad Rezaee , Jonas Baltrusaitis , Pedram Roghanchi

The paper is in general well written. It is easily readable and very interesting. The paper is very actual and the investigation is original. The paper is in line with the topic of the journal Minerals. From that point, I suggest the editor accept the paper for publication. In the manuscript, I did not mention any significant mistakes or lack, but there are some things that the authors need to correct so I recommend the acceptance but after minor revision.

My other comments are:

1.      Please give in the paper an explanation of why it was of importance to determine the specific surface area of the investigated samples.

2.      Please correct explanations for the FTIR measurements in order to be better connected with the results of the XRD.

3.      Please check some references which are in the text given as ERRORs

Best regards

Author Response

We appreciate your time and effort in reviewing our paper. we carefully applied your comments and address them in the manuscript. 

  1. Please give in the paper an explanation of why it was of importance to determine the specific surface area of the investigated samples.

Answer: Thank you for your comment. The specific surface area is important because is the area that is in contact with the simulated lung fluids in the dissolution experiments. The following paragraph explaining the importance was added to the paper:

Lines 435 to 443: “…The specific surface area and micro-pore analysis of the dust samples are important because they give the area that is in contact with the SLF in the dissolution experiments. The formation of the surface complexes is highly dependent on the exposed surface area and thus governs the dissolution process. The same mass (20 mg) was used for each dissolution experiment, therefore, a big difference in the specific surface area among the samples may influence the final dissolutions of the elements because it is presumed that a larger exposed area to the SLF can result in more dissolution and finally biased results. In this case, the specific surface area of the samples was very close to each other, representing the minimum influence of the exposed area in the dissolution experiments.”

  1. Please correct explanations for the FTIR measurements in order to be better connected with the results of the XRD.

Answer: Thank you for your recommendation. The reference “Silica” was changed for “Quartz” in the peak assignment table to make it easier to connect to the XRD results. Quartz and kaolinite were the only mineral species identified with the FTIR. The other functional groups correspond mainly to organic components that cannot be measured with the XRD, and thus, cannot be correlated.

  1. Please check some references which are in the text given as ERRORs

Answer: Thank you for your comment. We addressed your comment in the manuscript.

Reviewer 3 Report

Comments and Suggestions for Authors

Manuscript ID: minerals-1806727

Title: Characterization and Toxicity Analysis of Lab Created Respirable Coal Mine Dust from the Appalachians and Rocky Mountains Regions

Authors: Vanessa Salinas, Milton Das, Quiteria Jacquez, Alexandra Camacho, Katherine Zychowski, Mark Hovingh, Alexander Medina, Gayan Rubasinghege, Mohammad Rezaee, Jonas Baltrusaitis, Pedram Roghanchi

This paper represents a significant contribution to the literature and is well aligned with the Journal's scope. It should be accepted after moderate revisions. The following comments are offered to improve the overall quality and readability of the paper.

The specific suggestions are the following:

l  Abstract. Line 23 “….. may exhibit more toxicity than other geographic regions”, In this paper, the author studied the RCMD characteristics of Appalachians and Rocky Mountains Regions, the author should consider whether such a statement is reasonable, is it more scientific to express as “….. may exhibit more toxicity than Rocky Mountain?

l  Introduction. Why did you choose these two areas as the research object? Is there any scientific or social significance? Please add relevant introduction in this part, so that readers can have a clearer understanding of the research background.

l  Line 110 to 114, This paragraph is a conclusive statement, and the author should consider whether it is appropriate to appear in this position.

l  Materials and Methods. In this part, the author should give the basic information of the coal seams of studied coal mines, such as the quality information of the coal (moisture content, ash yield, volatile matter yield, coal rank), the age of coal formation, the thickness of the coal seams, etc. (added in Table 1).

l  Line 120 to 125, Is this part of the description necessary? Reviewers believe that this part has nothing to do with the content of the article, which will only increase readers' confusion, and suggest deleting or rewriting.

l  Line 130 to 131, and other sites, in this paper, "Error! Reference source not found" appears in large numbers. Please carefully check the submitted papers to avoid such unnecessary errors.

l  Line 135. “No. 6 (ASTM E11)”, which size it represents? Explain that it may be easier to read and understand.

l  Line 255-256. Why did A549 and T1 cells not choose to be exposed to medium concentrations of PM10?

l  Particle size distribution. Why did the author not use laser diffractometer device to analyze the particle size distribution of RCMD?

l  Line 335-336. It is more reasonable to calculate the relative mineral content by ash yield.

l  Figure 4. Please check this figure. Figure 4a and 4b have same elements (Ti, Sr, and Ba), but the concentrations are not coincident.

l  Line 426-427. Is it related to the selection of experimental samples?

l  Line 450-452. We didn't find the experimental results of this part.

l  Line 487. The modes of occurrences of elements in RCMD is also important in the dissolution, please consider the impact of this aspect.

l  Line 501-505, The results here have some contradictions with the pervious study (Sarver, 2019, Ref. 11), please discuss it.

l  Line 624. “IFN-γ, IL-10, IL12p70, IL-13, IL-4 and IL-6”, these nouns should be explained here to enhance the readability of the article.

l  The author discusses the properties of dust from many aspects, and whether these characteristics can be integrated and then discussed in combination with the toxicity characteristics?

Author Response

We appreciate your time and effort in reviewing our manuscript. We carefully addressed your comments in the paper. 
